# GCN2 eIF2 kinase promotes prostate cancer by maintaining amino acid homeostasis

**Ricardo A Cordova[1,2], Jagannath Misra[1], Parth H Amin[1], Anglea J Klunk[1], Nur P Damayanti[2,3], Kenneth R Carlson[1], Andrew J Elmendorf[1], Hyeong-Geug Kim[1], Emily T Mirek[4], Bennet D Elzey[5,6], Marcus J Miller[7], X Charlie Dong[1], Liang Cheng[2,8†], Tracy G Anthony[4], Roberto Pili[9]\*, Ronald C Wek[1,2]\*, Kirk A Staschke[1,2]\***

[1]Department of Biochemistry and Molecular Biology, Indiana University School of Medicine, Indianapolis, United States; [2]Indiana University Melvin and Bren Simon Comprehensive Cancer Center, Indianapolis, United States; [3]Department of Neurological Surgery, Indiana University School of Medicine, Indianapolis, United States; [4]Department of Nutritional Sciences, Rutgers University, New Brunswick, United States; [5]Department of Comparative Pathology, Purdue University, West Lafayette, United States; [6]Department of Urology, Indiana University School of Medicine, Indianapolis, United States; [7]Department of Medical and Molecular Genetics, Indiana University School of Medicine, Indianapolis, United States; [8]Department of Pathology and Laboratory Medicine, Indiana University School of Medicine, Indianapolis, United States; [9]Jacobs School of Medicine and Biomedical Sciences, University at Buffalo, Buffalo, United States

\*For correspondence:
rpili@buffalo.edu (RP);
rwek@iu.edu (RCW);
kastasch@iu.edu (KAS)

**Present address:** †Department of Pathology and Laboratory Medicine, Warren Alpert Medical School of Brown University, Providence, United States

**Abstract** A stress adaptation pathway termed the integrated stress response has been suggested to be active in many cancers including prostate cancer (PCa). Here, we demonstrate that the eIF2 kinase GCN2 is required for sustained growth in androgen-sensitive and castration-resistant models of PCa both in vitro and in vivo, and is active in PCa patient samples. Using RNA-seq transcriptome analysis and a CRISPR-based phenotypic screen, GCN2 was shown to regulate expression of over 60 solute-carrier (*SLC*) genes, including those involved in amino acid transport and loss of GCN2 function reduces amino acid import and levels. Addition of essential amino acids or expression of 4F2 (SLC3A2) partially restored growth following loss of GCN2, suggesting that GCN2 targeting of SLC transporters is required for amino acid homeostasis needed to sustain tumor growth. A small molecule inhibitor of GCN2 showed robust in vivo efficacy in androgen-sensitive and castration-resistant mouse models of PCa, supporting its therapeutic potential for the treatment of PCa.

## Editor's evaluation

This manuscript is an important body of work that addresses the role of the integrated stress response (ISR) and the role of the GCN2 protein kinase in prostate cancer. The studies comprehensively elucidate how GCN2 and amino acid transporters and uptake promote prostate cancer proliferation, as well as the therapeutic potential of inhibiting this pathway. This work, therefore, provides insights for both identification of new mechanisms and experimental therapeutics in prostate cancer.

**eLife digest** Prostate cancer is the fourth most common cancer worldwide, affecting over a million people each year. Existing drug treatments work by blocking the effects or reducing the levels of the hormone testosterone. However, these drug regimens are not always effective, so finding alternative treatments is an important area of research. One option is to target the 'integrated stress response', a pathway that acts as a genetic switch, turning on a group of genes that counteract cellular stress and are essential for the survival of cancer cells.

The reason cancer cells are under stress is because they are hungry. They need to make a lot of proteins and other metabolic intermediates to grow and divide, which means they need plenty of amino acids, the building blocks that make up proteins and fuel metabolism. Amino acids enter cells through molecular gates called amino acid transporters, and scientists think the integrated stress response might play a role in this process. One of the integrated stress response components is a protein called General Control Nonderepressible 2, or GCN2 for short. In healthy cells, this protein helps to boost amino acid levels when supplies start to run low.

Cordova et al. examined human prostate cancer cells to find out what role GCN2 plays in this cancer. In both lab-grown cells and tissue from patients, GCN2 was active and played a critical role in prostate tumor growth by turning on the genes for amino acid transporters to increase the levels of amino acids entering the cancer cells. Deleting the gene for GCN2, or blocking its effects with an experimental drug, slowed the growth of cultured prostate cancer cells and reduced tumor growth in mice. In these early experiments, Cordova et al. did not notice any toxic side effects to healthy tissues.

If GCN2 works in the same way in humans as it does in mice, blocking it might help to control prostate cancer growth. The integrated stress response is also active in other cancer types, so the same logic might apply to different tumors. However, before GCN2 blockers can become treatments, researchers need a more complete understanding of their molecular effects.

## Introduction

Prostate cancer (PCa) accounts for 20% of all newly diagnosed cancers and is the second leading cause of cancer-related deaths in men in the United States (*Siegel et al., 2019*). Many factors contribute to PCa, including genetics, diet, age, and race (*Gann, 2002*). The pathological grade of the PCa and whether it has disseminated into surrounding tissues is prognostic for risk and treatment efficacy (*Klusa et al., 2020*). Androgen receptor (AR) is considered the primary oncogenic driver for PCa (*Heinlein and Chang, 2004*; *Huggins and Hodges, 1941*), and therapies targeting AR activity are current standard of care, especially for those with metastatic PCa (*Polotti et al., 2017*; *Virgo et al., 2017*; *Watson et al., 2015*; *Crawford et al., 2019*). Unfortunately, patients develop resistance with time and fail therapy (*Watson et al., 2015*; *Bubley and Balk, 2017*). In addition, it remains unclear how additional pathways drive PCa through AR-dependent and AR-independent mechanisms. Therefore, the identification of novel targets for therapeutic intervention are critically needed and clinically relevant.

The integrated stress response (ISR) is critical for cell adaptation and survival to environmental stresses. Central to the ISR is phosphorylation of the α subunit of eIF2 (p-eIF2α), which triggers reductions in global protein synthesis to conserve energy and nutrients, and concomitantly facilitates adaptive gene expression (*Baird and Wek, 2012*; *Tian et al., 2021*). Four eIF2α kinases respond to distinct stress conditions: The eIF2α kinase GCN2 (EIF2AK4) is induced by amino acid starvation and other conditions that trigger ribosomal collisions or stalling (*Dong et al., 2000*; *Inglis et al., 2019*; *Wek et al., 1995*). PERK (EIF2AK3) is responsive to disruptions in the endoplasmic reticulum (*Walter and Ron, 2011*; *Wek and Cavener, 2007*). PKR (EIF2AK2) is activated by viral infection and inflammation (*Chukwurah et al., 2021*; *García et al., 2006*), and HRI (EIF2AK1) is enhanced by restricted heme or iron supply, oxidative stress, and disruptions in mitochondrial function (*Fessler et al., 2020*; *Guo et al., 2020*; *Suragani et al., 2012*). Activation of these ISR regulators may also occur in response to oncogenic drivers, such as MYC amplification and loss of the PTEN tumor suppressor, frequently observed in prostate tumors. Therefore, both extrinsic and intrinsic cell stresses can invoke the ISR in tumors and pharmacological inhibition of translational control is reported to trigger cytotoxicity in different cancer models (*Staschke and Wek, 2019*; *Tameire et al., 2019*; *Nguyen et al., 2018*).

To reprogram gene expression toward recovery from stress damage, the ISR also induces preferential translation of specific ISR gene transcripts, such as activating transcription factor 4 (ATF4), a transcriptional regulator of genes involved in amino acid metabolism and proteostasis control, oxidative stress defenses, along with feedback control of the ISR (*Harding et al., 1999*; *Young and Wek, 2016*; *Hinnebusch et al., 2016*). Therefore, p-eIF2α induces both translational and transcriptional modes of gene expression to achieve stress adaptation. Targeted inhibition of ATF4 impedes progression of MYC-driven cancers, emphasizing the important role of this ISR effector in tumor progression (*Staschke and Wek, 2019*; *Tameire et al., 2019*; *Nguyen et al., 2018*).

Here, we dissect the role of eIF2α kinase(s) induced in PCa and the mechanisms by which the ISR serves to promote growth of this cancer. Using androgen-sensitive and castration-resistant human PCa cell lines, primary human PCa tissues, and mouse xenograft models, we show that the ISR regulator GCN2 is activated in PCa. GCN2 induces expression of key ISR effector genes involved in nutrient transport that are critical for appropriate maintenance of free amino acid pools. Genetic or pharmacological inhibition of GCN2 dysregulates expression of these transporters and inhibits PCa growth. These results point to GCN2 as a potential therapeutic target for PCa.

## Results
### GCN2 activity supports PCa growth

To determine the role of individual eIF2α kinases and their importance for proliferation of PCa cells in culture, androgen-sensitive LNCaP cells cultured in androgen replete media conditions were individually depleted of HRI, PKR, PERK, or GCN2 using target-specific siRNAs. There was robust reduction of protein levels of each of the eIF2α kinases; however, only cells depleted for GCN2 or PKR showed a significant decrease in proliferation (*Figure 1A, B*, *Figure 1—figure supplement 1A*, and *Supplementary file 1*). Whereas GCN2 depletion lowered p-eIF2α levels and ATF4 expression, knockdown of PKR, HRI, or PERK had no effect on p-eIF2α and ATF4 protein levels (*Figure 1B* and *Figure 1—figure supplement 1A*). These results suggest that LNCaP cell proliferation requires GCN2 as an ISR regulator.

To broaden our investigation to other PCa cell lines, we knocked down GCN2 in the androgen-sensitive line LAPC-4 and in the castration-resistant cell lines C4-2B, MR49F, 22Rv1, and PC-3. MR49F cells express AR, but are resistant to enzalutamide (*Kuruma et al., 2013*), 22Rv1 cells express the ARv7 splice variant and are also resistant to enzalutamide (*DeSantis et al., 2016*; *Li et al., 2013*; *Sramkoski et al., 1999*), PC-3 is AR null (*Tilley et al., 1990*). In each of these PCa cell lines, depletion of GCN2 reduced proliferation and lowered ATF4 expression (*Figure 1C*, *Figure 1—figure supplement 1B*, and *Supplementary file 1*). In 22Rv1 and PC-3 cells, we were also able to achieve a deletion of *EIF2AK4* (GCN2) using CRISPR/Cas9, which reduced p-eIF2α and ATF4 levels (*Figure 1—figure supplement 2A*). In the case of 22Rv1, we determined a similar slow growth phenotype as observed with siRNA knockdown of GCN2 (*Figure 1—figure supplement 2B* and *Supplementary file 1*). Reintroduction of GCN2 into the GCN2 KO cells rescued the slow growth phenotype, and expression of ATF4 and its target gene ASNS (*Figure 1—figure supplement 2B*, *Figure 1—figure supplement 2C*, and *Supplementary file 1*). The small molecule GCN2iB is a selective inhibitor of GCN2 eIF2α kinase activity (*Nakamura et al., 2018*). Using the inhibitory concentration range of 500 nM to 10 µM of GCN2iB, we determined that pharmacological inhibition of GCN2 also caused significant growth inhibition of LNCaP cells, and similarly reduced the growth of C4-2B, 22Rv1, and PC-3 cells (*Figure 1D*, *Figure 1—figure supplement 3A*, and *Supplementary file 1*). Treatment of GCN2 KO cells with up to 10 µM GCN2iB had no effect on growth demonstrating the specificity of this inhibitor (*Figure 1—figure supplement 3B* and *Supplementary file 1*). As expected, treatment of the LNCaP, 22Rv1, C4-2B, or PC-3 cells with GCN2iB lowered activation of GCN2 as measured by autophosphorylation of the activation loop (p-GCN2-T899), reduced p-eIF2α levels, and lowered expression of ATF4, and its target genes ASNS and TRIB3. Importantly, treatment with GCN2iB did not alter protein levels of AR (*Figure 1E* and *Figure 1—figure supplement 3C*). To further test the importance of the ISR in PCa, we selectively depleted ATF4 using siRNA in LNCaP, MR49F, C4-2B, and PC-3 cells and observed similar growth inhibition for each cell line (*Figure 1—figure supplement 1B*, *Figure 1—figure supplement 4*, and *Supplementary file 1*). Importantly, GCN2 is not active in the non-cancerous prostate epithelial cell line BPH-1 (*Hayward et al., 1995*), and loss of GCN2 or

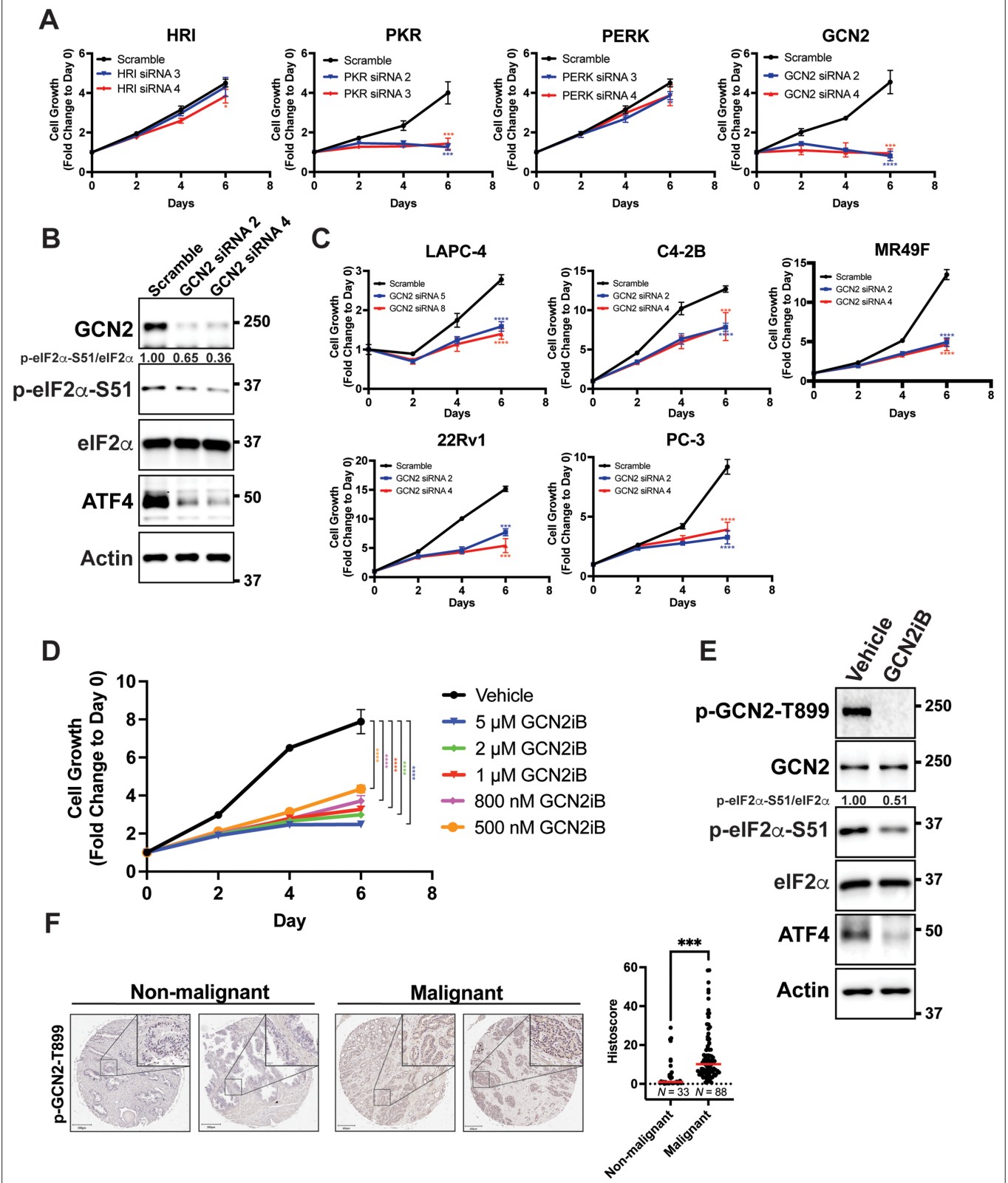

**Figure 1.** GCN2 promotes growth of prostate cancer (PCa) cells. (**A**) Expression of the indicated eIF2α kinase was reduced in LNCaP cells using gene-specific siRNAs. Two different siRNAs were used for knockdown of each eIF2α kinase and compared to scrambled siRNA control. Cell growth was measured in replicate wells ($N = 5$) for up to 6 days and is plotted as fold change (mean ± standard deviation [SD]) relative to day 0. Statistical significance was determined using a two-way analysis of variance (ANOVA) as described in ***Supplementary file 1***; *p ≤ 0.05, ***p ≤ 0.001, ****p ≤

*Figure 1 continued on next page*

*Figure 1 continued*

0.0001. (**B**) LNCaP cells were transfected with two different siRNAs targeting GCN2 or a scramble siRNA control and cell lysates were prepared and immunoblotted for the indicated proteins. Molecular weight markers are shown in kilodaltons. The relative levels of p-eIF2α normalized to total eIF2α compared to scramble siRNA control are indicated. (**C**) Expression of GCN2 was knocked-down in LAPC-4, C4-2B, MR49F, 22Rv1, or PC-3 cells using two different siRNAs and compared to scrambled siRNA control. Cell growth was measured for up to 6 days in replicate wells (*N* = 5) as described in **A**. Statistical significance was determined using a two-way ANOVA as described in *Supplementary file 1*; ***p ≤ 0.001, ****p ≤ 0.0001. (**D**) LNCaP cells were treated with indicated concentrations of GCN2iB and cell growth was measured for up to 6 days in replicate wells (*N* = 5) as described in **A**. Statistical significance was determined using a two-way ANOVA as described in *Supplementary file 1*; ****p ≤0.0001. (**E**) LNCaP cells were treated with GCN2iB (2 µM) or DMSO control for 24 hr and protein lysates were analyzed by immunoblot using antibodies that recognize total or phosphorylated GCN2-T899, total or phosphorylated eIF2α–S51, ATF4, or actin as indicated. Relative levels of p-eIF2α normalized to total eIF2α are shown. (**F**) Levels of p-GCN2 were measured in prostate tumor microarrays (Biomax PR1921b and PR807c) using immunohistochemistry (IHC). Staining for p-GCN2-T899 from non-malignant (*N* = 33) and malignant PCa tissue (*N* = 88) from patients >50 years old was analyzed and quantified using QuPath to determine the histoscore and is represented as a scatterplot. Statistical significance was determined using an unpaired two-tailed *t*-test; *p ≤ 0.05. Representative images showing p-GCN2-T899 staining of normal and malignant prostate tissues are shown. Scale bars shown are 200 µm (main image) and 20 µm (insert).

The online version of this article includes the following figure supplement(s) for figure 1:

**Figure supplement 1.** Knockdown of eIF2α kinases and ATF4 in prostate cancer (PCa) cell lines.

**Figure supplement 2.** CRISPR/Cas9 knockout of GCN2 in 22Rv1 and PC-3 cells.

**Figure supplement 3.** Inhibition of the integrated stress response (ISR) by GCN2iB in prostate cancer (PCa) cell lines.

**Figure supplement 4.** Knockdown of ATF4 reduces growth in prostate cancer (PCa) cell lines.

**Figure supplement 5.** Inhibition of GCN2 in non-cancerous prostate cell line BPH-1.

**Figure supplement 6.** p-GCN2 staining of prostate core needle specimens.

ATF4, or pharmacological inhibition of GCN2, did not reduce p-eIF2α or ATF4 levels, or significantly impact the growth of these cells (*Figure 1—figure supplement 5A*, *Figure 1—figure supplement 5B*, *Figure 1—figure supplement 5C*, and *Supplementary file 1*). These results support the hypothesis that the eIF2α kinase activity of GCN2 and the ISR support PCa proliferation.

We next sought to determine whether GCN2 is activated in human malignant PCa tissues. Using immunohistochemistry and prostate tumor microarrays featuring normal and non-malignant prostate tissue (*N* = 33) and malignant tumor tissues (*N* = 88), we determined that there was enhanced GCN2 activity in the tumor samples compared to normal tissues as measured by p-GCN2-T899 staining (*Figure 1F*). In addition, increased p-GCN2 (Thr-899) staining was also observed in prostate tumors from patient core needle biopsies as compared to adjacent non-malignant tissue (*Figure 1—figure supplement 6*). Collectively, these results indicate that GCN2 is active basally in both androgen-sensitive and castration-resistant PCa models without appreciable extrinsic stress treatments, confers growth advantages, and is activated in PCa patient tissues.

## GCN2 induces ISR transcriptome featuring expression of amino acid transporters

To address the mechanisms by which GCN2 confers growth advantages in PCa, we cultured LNCaP cells in the presence or absence of GCN2iB and then measured changes in the transcriptome by RNA-seq. With 24 hr of GCN2iB treatment, 239 transcripts showed a significant ≥twofold increase, whereas 374 transcripts were decreased ≥twofold (fdr ≤0.05) compared to vehicle-treated cells (*Figure 2A*, *Supplementary file 5*). We carried out a gene set enrichment analysis (GSEA) of the differentially expressed transcripts to distinguish the molecular functions that were altered by GCN2 inhibition. Included among the functional gene groups showing significant dependence on GCN2 for expression were those linked to amino acid deprivation and transport (*Figure 2B, C*). These genes are representative of ISR-targeted gene groups and include genes regulated by ATF4 such as *DDIT3 (CHOP)*, *ASNS*, and *TRIB3*, those involved in glycine–serine biosynthesis (*PHGDH*, *PSAT1*, *PSPH*, and *SHMT2*), aminoacyl-tRNA synthetases, and a large number of SLC transporters, many of which function in the transport of amino acids (*Figure 2B* and *Supplementary file 2*). As illustrated in the heat map, expression of the *SLC* genes was significantly lowered after 6 hr of GCN2iB treatment, with further reductions following 24 hr (*Figure 2C*). It is noteworthy that of the 61 known *SLC* genes involved in the transport

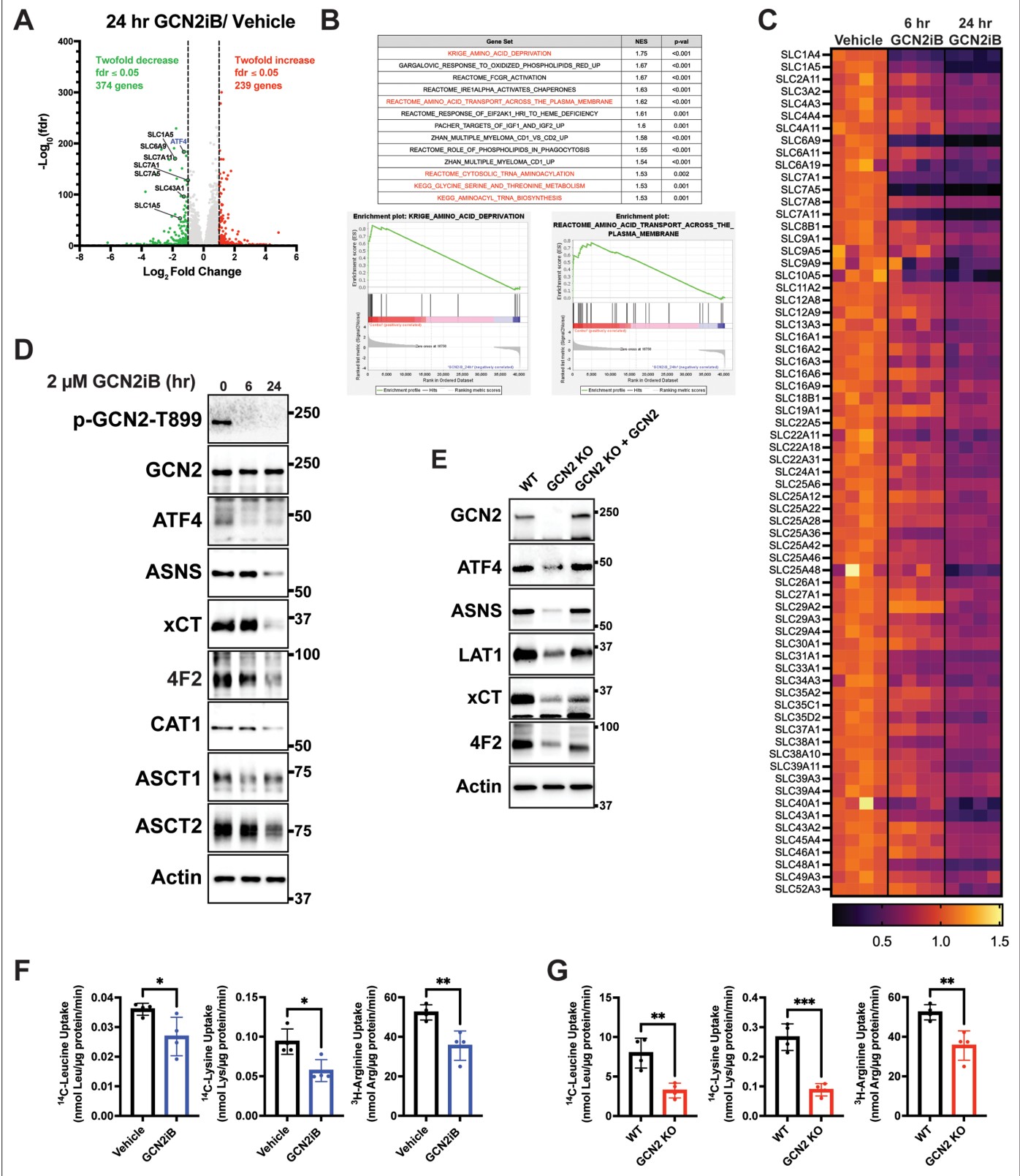

**Figure 2.** GCN2 induces integrated stress response (ISR) transcriptome featuring expression of amino acid transporters. (**A**) Volcano plot illustrating log₂ fold change in gene transcript levels with adjusted p value (−log₁₀) comparing LNCaP cells treated with GCN2iB (2 μM) versus vehicle control (DMSO) for 24 hr. Several amino acid transporters reduced by GCN2iB treatment are highlighted. (**B**) Plots from gene set enrichment analysis (GSEA) of gene expression in LNCaP cells treated with GCN2iB (2 μM) for 24 hr versus vehicle control. (**C**) Heat map displaying significantly downregulated *SLC* genes

*Figure 2 continued on next page*

*Figure 2 continued*

as indicated in panel **A**. The heat map compares gene transcript levels from LNCaP cells treated with vehicle (DMSO), or GCN2iB (2 µM) for 6 or 24 hr. Four biological replicates were measured for each treatment group. Transcript levels (normalized read counts) are shown relative to the average of the vehicle control samples for each gene. (**D**) Lysates were prepared from LNCaP cells treated with 2 µM GCN2iB or vehicle control (DMSO) for 6 or 24 hr and immunoblot analysis were carried out using antibodies that recognize ATF4, ASNS, xCT (SLC7A11), 4F2 (SLC3A2), CAT1 (SLC7A1), ASCT1 (SLC1A4), ASCT2 (SLC1A5), or actin. Molecular weight markers are indicated in kilodaltons. (**E**) 22Rv1 WT cells, 22Rv1 GCN2 KO cells, and 22Rv1 GCN2 KO complemented with GCN2 cells were cultured for 24 hr. Lysates were prepared and analyzed by immunoblot for the indicated proteins. (**F**) Amino acid uptake measurements in LNCaP and 22Rv1 cells treated with vehicle (DMSO) or GCN2iB (2 µM) for 24 hr. (**G**) Amino acid uptake measurements for 22Rv1 WT or 22Rv1 GCN2 KO cells cultured for 24 hr. Statistical significance was determined using an unpaired two-tailed *t*-test (*N* = 4); *p ≤ 0.05, **p ≤ 0.01, ***p ≤ 0.001.

The online version of this article includes the following figure supplement(s) for figure 2:

**Figure supplement 1.** Inhibition of GCN2 in prostate cancer (PCa) cells reduces expression of *SLC* genes involved in amino acid transport.

of amino acids, 27 were measurably expressed in LNCaP cells, and 20 of these expressed transcripts were significantly decreased following treatment with GCN2iB (*Figure 2B* and *Supplementary file 2*).

Lowered expression of SLC proteins upon GCN2 inhibition was further validated by immunoblot analyses. xCT (SLC7A11), CAT-1 (SLC7A1), ASCT1 (SLC1A4), ASCT2 (SLC1A5), and 4F2 (SLC3A2) protein levels were sharply lowered with GCN2iB treatment of LNCaP cells, along with reduced expression of ATF4 and its target gene ASNS (*Figure 2D* and *Figure 2—figure supplement 1A*). Similarly, treatment of C4-2B, 22Rv1, and PC-3 cells with GCN2iB also reduced LAT-1 (SLC7A5), xCT (SLC7A11), and 4F2 (SLC3A2) (*Figure 1—figure supplement 3C*). We also measured transporter proteins in 22Rv1 and PC-3 cells and showed that deletion of *EIF2AK4* (GCN2) also diminished protein levels of xCT (SLC7A11), LAT-1 (SLC7A5) (LAT1), and 4F2 (SLC3A2) (*Figure 2E*, *Figure 2—figure supplement 1B*). Furthermore, the reintroduction of GCN2 fully restored expression of ATF4 and ASNS and partially restored expression of all three SLC proteins in 22Rv1 cells (*Figure 2E*). To more directly address the effects of GCN2 inhibition on amino acid transport, we measured uptake of leucine, lysine, and arginine in LNCaP and 22Rv1 cells. Treatment of LNCaP cells with GCN2iB or deletion of GCN2 in 22Rv1 cells significantly lowered the uptake of each of these amino acids (*Figure 2F, G*), suggesting that GCN2-directed gene expression is important for uptake of amino acids in PCa cells.

## GCN2 is critical for maintenance of free amino acid pools

Given that GCN2 is important for expression of *SLC* genes and amino acid uptake, we next addressed whether the intracellular levels of free amino acids are disrupted upon inhibition of GCN2. LNCaP cells were cultured in the presence or absence of GCN2iB for 8 hr and free amino acid levels were measured. With pharmacological inhibition of GCN2, there was a sharp reduction in the amounts of essential amino acids (EAA), including histidine, the branched-chain amino acids, threonine, phenylalanine, and methionine, and non-essential amino acids (NEAA), such as glutamate and glutamine, aspartate, asparagine, alanine, and tyrosine (*Figure 3A*). We next considered whether supplementing the culture medium with additional amino acids would alleviate the growth defect that occurs with the inhibition of GCN2. While GCN2iB treatment of LNCaP cells cultured in standard medium markedly reduced growth, supplementation of additional EAAs into the medium reversed this proliferation defect and increased free amino acid pools in treated cells (*Figure 3B*, *Figure 3—figure supplement 1*, *Figure 3—figure supplement 2*, and *Supplementary file 1*). By contrast, supplementation with additional NEAA did not suppress the LNCaP growth defect triggered by GCN2iB treatment (*Figure 3—figure supplement 2* and *Supplementary file 1*).

To address the effect of GCN2 inhibition and EAA supplementation on the cell cycle in cultured PCa cells, LNCaP cells were treated with GCN2iB or GCN2 expression was reduced with siRNA. Both methods of GCN2 depletion caused an increase in the G1 phase of the cell cycle concomitant with a reduction in S phase, indicative of checkpoint control of the cell cycle that can occur with amino acid depletion (*Lehman et al., 2015*; *Leung-Pineda et al., 2004*; *Figure 3C* and *Figure 3—figure supplement 3A*). Supplementation of LNCaP cells with EAA reversed the accumulation of cells in G1 phase elicited by GCN2iB (*Figure 3C*). These results suggest that inhibition of GCN2 can cause cell cycle arrest, which was recapitulated with siRNA knockdown of GCN2 or ATF4 (*Figure 3—figure supplement 3A*). Similar findings were observed in 22Rv1 cells deleted for GCN2 or in other PCa cell

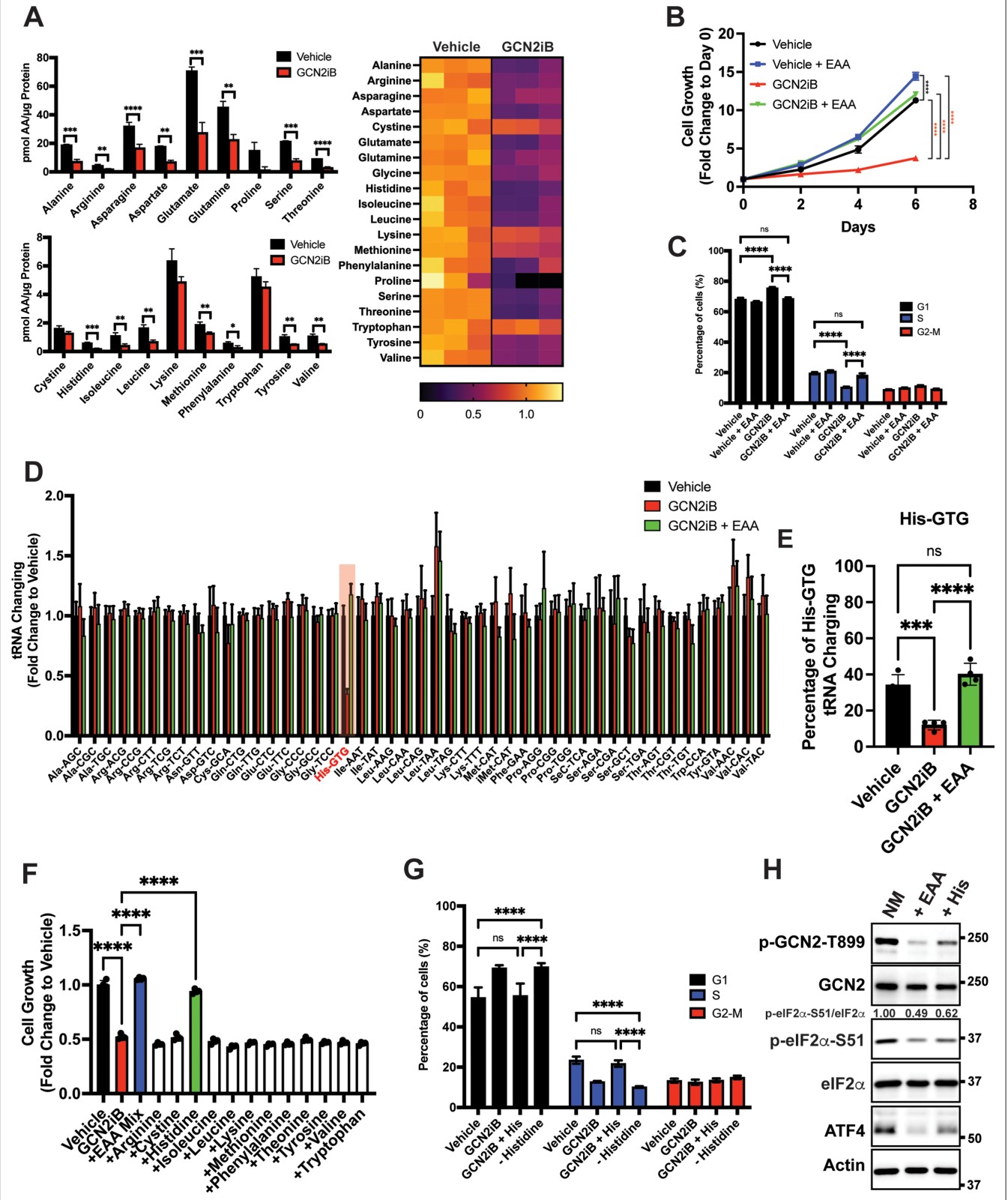

**Figure 3.** GCN2 is critical for maintenance of free amino acids. (**A**) Amino acid measurements of LNCaP cells treated with 2 μM GCN2iB or vehicle control (DMSO) for 8 hr. Bar graphs in the top panel show high abundance amino acids and the lower panel those with lower levels. The heat map on the right shows fold change in amino acid abundance for each biological replicate of GCN2iB-treated LNCaP cells versus the vehicle with the scale showing the highest fold change in yellow and lowest in purple. Statistical significance was determined using an unpaired two-tailed *t*-test. Error bars

*Figure 3 continued on next page*

*Figure 3 continued*

indicate standard deviation (SD) (N = 3); *p≤0.05, **p≤0.01, ***p ≤ 0.001, ****p≤0.0001. (**B**) LNCaP cells were treated with vehicle, GCN2iB (2 μM), vehicle + essential amino acids (EAA), or GCN2iB (2 μM) + EAA, and cell growth was measured for up to 6 days. Error bars indicate SD (N = 5). Statistical significance was determined using a two-way analysis of variance (ANOVA) as described in *Supplementary file 1*; ****p ≤ 0.0001. (**C**) Cell cycle analyses of LNCaP cells treated with vehicle, GCN2iB (2 μM), vehicle + EAA, or GCN2iB (2 μM) + EAA for 48 hr. Statistical significance was determined using a one-way ANOVA with Tukey's multiple comparisons. Error bars indicate SD (N = 3); ***p≤0.001, ****p≤0.0001. (**D**) Genome-wide tRNA charging analysis (CHARGE-seq) of LNCaP cells treated with vehicle (DMSO), GCN2iB (2 μM), or GCN2iB (2 μM) + EAA for 8 hr. The tRNA charging ratio is shown as a bar graph with fold change compared to vehicle. Only tRNA isoacceptors measured in LNCaP cells are shown. Error bars indicate SD (N = 4). (**E**) tRNA charging percentage for tRNA^His in LNCaP cells treated with vehicle, GCN2iB, or GCN2iB + EAA. Statistical significance was determine using a one-way ANOVA with Tukey's multiple comparisons (N = 4); ***p≤0.001, ****p≤0.0001. (**F**) LNCaP cells were treated with vehicle, GCN2iB (2 μM), GCN2iB (2 μM) + EAA, or GCN2iB (2 μM) combined with the indicated individual amino acids. Cell growth was measured at 4 days in triplicate wells (N = 3). Statistical significance was determined using a one-way ANOVA with Tukey's multiple comparisons. Error bars indicate SD; ****p≤0.0001. (**G**) Cell cycle analysis of LNCaP cells were treated with vehicle, GCN2iB (2 μM), GCN2iB (2 μM) + histidine (200 μM), or with media lacking histidine for 48 hr. Statistical significance was determined using a one-way ANOVA with Tukey's multiple comparisons. Error bars indicate SD (N = 3); ***p≤0.001, ****p ≤0.0001. (**H**) LNCaP cells were cultured in normal media, media supplemented with EAA mix, or media supplemented with histidine (200 μM) for 24 hr. Lysates were analyzed by Immunoblot using antibodies that recognize total or phosphorylated GCN2-T899, total or phosphorylated eIF2α−S51, ATF4, or actin. Molecular weight markers are presented in kilodaltons for each immunoblot panel. The relative levels of p-eIF2α normalized to total eIF2α compared to normal media (NM) control are indicated.

The online version of this article includes the following figure supplement(s) for figure 3:

**Figure supplement 1.** Amino acid measurements of LNCaP cells treated with vehicle (DMSO), 2 μM GCN2iB, or 2 μM GCN2iB + essential amino acid (EAA) for 48 hr.

**Figure supplement 2.** Effect of AA supplementation on growth of LNCaP cells.

**Figure supplement 3.** Loss of GCN2 or ATF4 expression in prostate cancer (PCa) cell lines induces cell cycle arrest.

**Figure supplement 4.** GCN2 inhibition and supplementation with essential amino acids (EAA) affect charging of tRNA^HIS and global translation.

**Figure supplement 5.** Growth of MR49F and 22Rv1 can be rescued with amino acid supplements.

lines when GCN2 or ATF4 levels were reduced with siRNA (*Figure 3—figure supplement 3A* and *Figure 3—figure supplement 3B*).

Accumulating uncharged tRNA occurs upon severe depletion of cognate amino acids and is a primary signal activating GCN2 (*Battu et al., 2017*). We cultured LNCaP cells in the presence or absence of GCN2iB and measured tRNA charging genome-wide using CHARGE-seq (*Pavlova et al., 2020*). There was a reduction of aminoacylated tRNA^His upon GCN2iB treatment that was restored when additional EAA were supplemented to the growth medium (*Figure 3D, E*, and *Supplementary file 3*). These results were further supported by measurements of tRNA^His charging levels by qRT-PCR. Addition of EAA to cultured LNCaP cells significantly increased tRNA^His charging levels while treatment with GCN2iB dramatically reduced tRNA^His charging levels which was reversed by the addition of EAA (*Figure 3—figure supplement 4A*). These results support the premise that histidine depletion and the accompanying accumulation of uncharged tRNA^His can lead to activation of GCN2. Upon treatment with GCN2iB, which lowered p-eIF2α and blocked appropriate induction of the ISR, there was severe depletion of histidine accompanied by accumulation of uncharged tRNA^His, culminating in a slow growth phenotype.

To further address the role of histidine availability in the slow growth phenotype associated with GCN2 inhibition, we cultured GCN2iB-treated LNCaP cells in medium with added supplements of individual amino acids. In alignment with our charged tRNA^His measurements, only the addition of histidine to the culture medium reversed the growth inhibition triggered by GCN2iB (*Figure 3F* and *Figure 3—figure supplement 2*). Removal of histidine from the growth medium increased G1 accumulation similar to GCN2iB treatment, while supplementation of additional histidine to the medium alleviated the cell cycle arrest triggered by GCN2iB treatment (*Figure 3G*). Finally, supplementation with the full complement of EAA or histidine alone significantly lowered p-GCN2, p-eIF2α, and ATF4 expression, although it is noted that histidine supplementation rendered only partial suppression of the ISR compared to addition of EAA (*Figure 3H*). Accompanying this reduction in activation of GCN2, supplementation with histidine sharply increased aminoacylation of tRNA^His (*Figure 3E*) and there was a trend toward enhanced global protein synthesis as measured by puromycin incorporation. It is curious that there was a trend toward decreased bulk translation upon GCN2iB treatment that was restored with the addition of histidine (*Figure 3—figure supplement 4B*). As noted above,

GCN2 inhibition for 48 hr adversely affected the cell cycle and amino acid levels, which together are suggested to reduce bulk translation upon GCN2iB treatment despite lowered p-eIF2α levels.

We next determined whether EAA supplementation in other PCa cell lines also suppressed the slow growth phenotype triggered by loss of GCN2 activity. Addition of EAA or histidine alone also suppressed the slow growth phenotype of MR49F cells treated with GCN2iB (*Figure 3—figure supplement 5A*). However, in 22Rv1 cells, EAA or lysine alleviated the slow growth associated with deletion of *EIF2AK4* (GCN2) (*Figure 3—figure supplement 5B*). These growth supplementation studies suggest that while GCN2 is indispensable for maintaining EAA in PCa cells, the precise processes by which GCN2 carries out these functions can vary among PCa cell lines presumably due to differences in their unique genetic features and cell-line-specific buffering mechanisms accompanying specific gene expression and metabolic programs. Depletion of GCN2 in LNCaP and MR49F cells renders them vulnerable to histidine depletion, whereas deletion of GCN2 sensitizes 22Rv1 cells to lysine depletion.

## Select GCN2-regulated SLC transporters are critical for PCa fitness

Given that GCN2 has a primary role for inducing expression of *SLC* genes, we sought to determine the contributions of these transporters for PCa growth and viability. To evaluate their contributions in an unbiased manner, we utilized a previously reported CRISPR/Cas9 library targeting 394 human SLC genes and pseudogenes (*Girardi et al., 2020*). Each of the SLC genes was targeted with six specific single guide RNAs (sgRNAs), along with negative controls for target specificity and positive controls for assessing cell fitness. The sgRNA library was stably expressed in LNCaP and 22Rv1 cells using lentivirus delivery. Cells were then subjected to puromycin selection and then cultured for 3–4 weeks (15–20 population doublings) as previously reported (*Girardi et al., 2020*). Genomic DNA (gDNA) was prepared and encoded sgRNA sequences were amplified by PCR, sequenced, and their composition was compared to the starting cells with the sgRNA library at day 0. Depletion of the cells expressing specific sgRNAs were then evaluated for statistical significance, and cell fitness was scored and ranked. A collection of SLC genes showed significantly diminished growth, including several that require GCN2 for expression, such as *LAT1 (SLC7A5)*, *SLC25A36*, *SLC25A46*, *SLC27A1*, *SLC35A2*, and *4F2 (SLC3A2)* (*Figure 4A* and *Supplementary file 4*). *SLC27A1* and *4F2 (SLC3A2)* were required for efficient growth of both LNCaP and 22Rv1 cells (*Figure 4B* and *Figure 4—figure supplement 1*). These results suggest that PCa cells require a specific collection of *SLC* genes for growth and a subset of these are dependent on GCN2 for full expression.

Given the importance of 4F2 (SLC3A2) in the transport of bulk amino acids (*Pizzagalli et al., 2021*), we next focused on the GCN2-dependent regulation of this transport regulator. Treatment of LNCaP cells with the GCN2iB for 6 or 24 hr resulted in a reduction in GCN2 activation, as shown by p-GCN2, and lowered ATF4 mRNA and protein. Levels of 4F2 (SLC3A2) protein and mRNA were also lowered with inhibition of GCN2, although this reduction occurred after 24 hr of treatment (*Figure 4C* and *Figure 4—figure supplement 2A*). Consistent with our findings that histidine limitation in part drives GCN2 activation in LNCaP cells, addition of histidine to the growth media of LNCaP cells in culture reduced GCN2 activation and lowered expression of ATF4 and 4F2 (SLC3A2) protein and mRNA, while removing histidine from the media further increased GCN2 activation and expression of ATF4 and 4F2 (SLC3A2) (*Figure 4D* and *Figure 4—figure supplement 2B*). Salubrinal has been reported to increase p-eIF2α by inhibiting its PP1-directed dephosphorylation (*Boyce et al., 2005*). Supporting the idea that p-eIF2α is critical for 4F2 (SLC3A2) expression, addition of salubrinal alone increased 4F2 (SLC3A2) expression and also restored 4F2 (SLC3A2) expression in cells treated with GCN2iB (*Figure 4—figure supplement 3A*). In addition, overexpression of the PP1-targeting protein GADD34 (*Novoa et al., 2001*) reduced phosphorylation of eIF2α and expression of ATF4 and 4F2 (SLC3A2) (*Figure 4—figure supplement 3B*). Further supporting the central role of activation of GCN2 in the induction of 4F2 (SLC3A2) expression, treatment of LNCaP cells with halofuginone, an inhibitor of EPRS and prolyl-tRNA charging (*Keller et al., 2012*) resulted in robust activation of GCN2 and increased expression of ATF4 and 4F2 (SLC3A2) protein and mRNA (*Figure 4E* and *Figure 4—figure supplement 2C*). These results support the model that activation of GCN2 during nutrient stress is a major inducer of 4F2 (SLC3A2) expression.

We next addressed whether 4F2 (SLC3A2) expression is linked to the growth phenotype following loss of GCN2. First, knockdown of 4F2 (SLC3A2) expression using siRNA significantly reduced growth

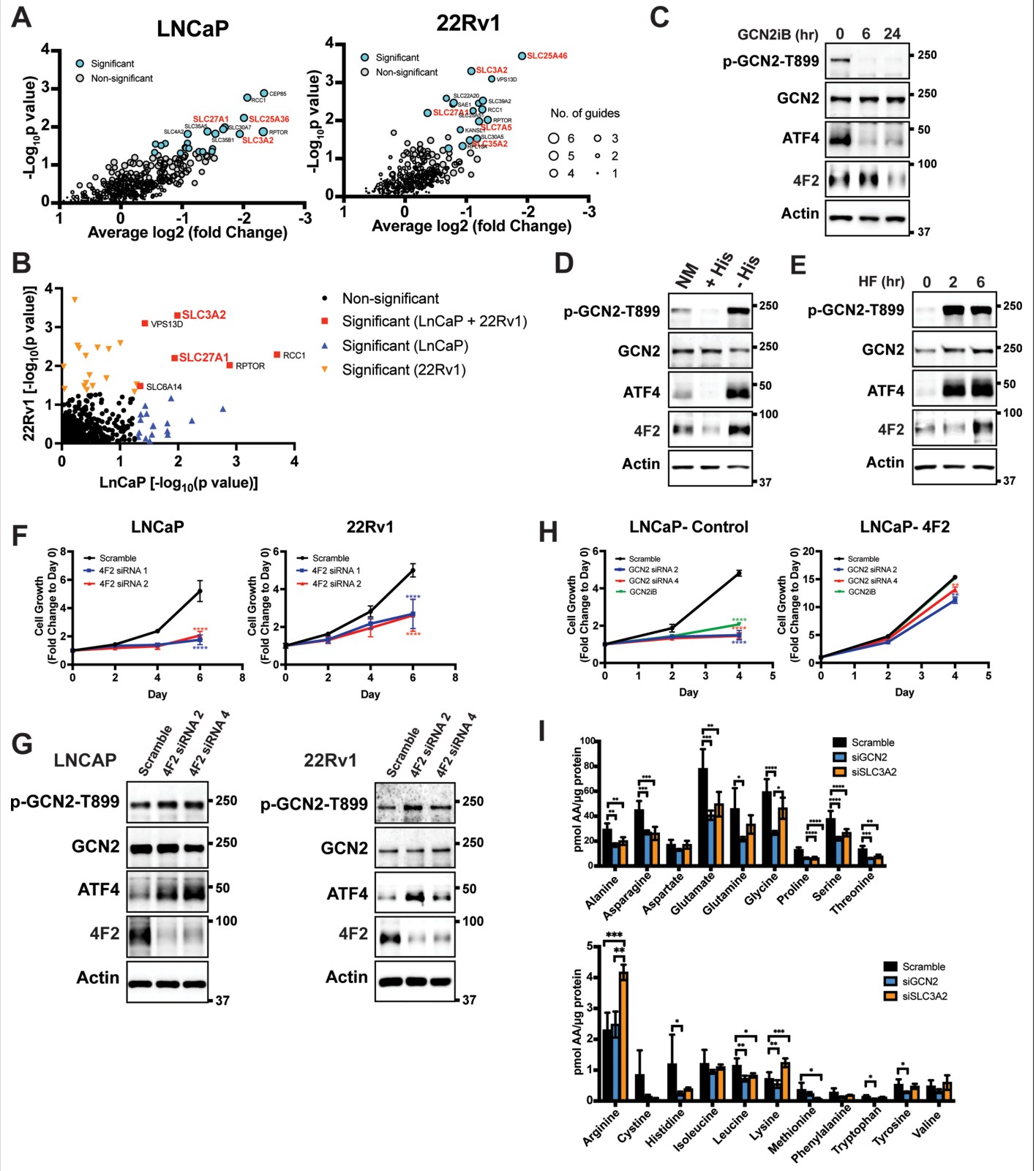

**Figure 4.** GCN2 is critical for expression of 4F2 (SLC3A2) and facilitates amino acid transport. (**A**) Gene-level depletion for LNCaP and 22Rv1 cells. The average log2 fold change for the single guide RNAs (sgRNAs) for each gene is shown on the x-axis. Significantly depleted genes (p ≤ 0.05) in LNCaP or 22Rv1 are indicated. Circle size indicates the number of significant sgRNAs. *SLC* genes in red are dependent on GCN2 for expression. (**B**) Plot of −Log₁₀(p value) for depleted genes identified in CRISPR screen for LNCaP versus 22Rv1 cells. Significantly depleted genes (p ≤ 0.05) in LNCaP, 22Rv1

*Figure 4 continued on next page*

*Figure 4 continued*

or both cell lines are indicated. SLC genes in red are GCN2 dependent. (**C**) Lysates from LNCaP cells were treated with 2 μM GCN2iB for 6 or 24 hr, or with vehicle (DMSO) were analyzed by immunoblot analyses using antibodies that recognize total or phosphorylated GCN2-T899, ATF4, 4F2 (SLC3A2), or actin. Molecular weight markers are indicated in kilodaltons for the panels. (**D**) LNCaP cells were cultured in standard culture conditions (NM: normal media), media supplemented with 200 μM histidine (+His), or media depleted of histidine (−His) for 24 hr. Lysates were analyzed by immunoblot analyses using antibodies that recognize total or phosphorylated GCN2-T899, ATF4, 4F2 (SLC3A2), or actin. (**E**) LNCaP cells were treated with 100 nM halofuginone (HF) for 2 and 6 hr or vehicle (DMSO). Lysates were analyzed by Immunoblot using antibodies that recognize the indicated proteins. (**F**) 4F2 (SLC3A2) expression was reduced in LNCaP or 22Rv1 cells using two different siRNAs or scramble siRNA as a control. Cell growth was measured in replicate wells ($N = 5$) for up to 6 days and are plotted relative to day 0 (mean ± standard deviation [SD]). Statistical significance was determined using a two-way analysis of variance (ANOVA) as described in *Supplementary file 1*; ****$p \leq 0.0001$. (**G**) LNCaP cells transfected with two different siRNAs targeting 4F2 (SLC3A2) or scramble siRNA for 48 hr. Lysate was prepared and analyzed by immunoblot using antibodies that recognize total or phosphorylated GCN2-T899, total or phosphorylated eIF2α–S51, ATF4, 4F2 (SLC3A2), or actin. (**H**) LNCaP cells stably overexpressing SLC3CA2 or vector control were transfected with two different siRNAs targeting GCN2 or scrambled control. Cells were then treated with GCN2iB (2 μM) or vehicle and growth was measured in replicate wells ($N = 5$) and is plotted relative to day 0 (mean ± SD). Statistical significance was determined using a two-way ANOVA as described in *Supplementary file 1*; **$p \leq 0.01$, ****$p \leq 0.0001$. (**I**) Amino acid measurements of LNCaP cells transfected siRNA targeting GCN2 ($N = 4$), 4F2 (SLC3A2, $N = 4$), or scramble control ($N = 8$). Two separate bar graphs show high abundance (top) and low abundance (bottom) amino acids. Statistical significance was determined using a two-way ANOVA as described in *Supplementary file 1*. Error bars indicate SD; *$p \leq 0.05$; **$p \leq 0.01$; ***$p \leq 0.001$; ****$p \leq 0.0001$.

The online version of this article includes the following figure supplement(s) for figure 4:

**Figure supplement 1.** SLC-specific CRISPR/Cas9 KO library reveals that 4F2 (SLC3A2) is critical for cell fitness in LNCaP and 22Rv1 cells.

**Figure supplement 2.** GCN2 regulates 4F2 (SLC3A2) mRNA levels in LNCaP cells.

**Figure supplement 3.** GCN2 and the integrated stress response (ISR) control expression of 4F2 (SLC3A2) which provides a growth advantage in prostate cancer (PCa) cell lines.

in LNCaP and 22Rv1 cells and led to activation of GCN2 and increased ATF4 protein levels (*Figure 4F, G*, and *Supplementary file 1*). Importantly, knockdown of 4F2 (SLC3A2) expression did not impact growth of non-cancerous BPH-1 cells (*Figure 1—figure supplement 5B* and *Figure 1—figure supplement 5C*). As was shown earlier, loss of GCN2 function using siRNA or GCN2iB treatment in cultured LNCaP cells reduced growth (*Figure 1A, D*). This growth defect was rescued by overexpression of 4F2 (SLC3A2) (*Figure 4H* and *Supplementary file 1*). Importantly, overexpression of 4F2 (SLC3A2) reduced GCN2 activity, which lowered p-GCN2 and p-eIF2α levels, along with ATF4 protein expression, and provided a growth advantage in both wild-type LNCaP and 22Rv1 cells (*Figure 4—figure supplement 3C* and *Figure 4—figure supplement 3D*). We next tested the impact of 4F2 (SLC3A2) knockdown on intracellular amino acid levels. As anticipated, knockdown of SLC3A2 reduced a subset of amino acids as compared to knockdown of GCN2 (*Figure 4I*). These results suggest that expression of *SLC* genes, especially 4F2 (SLC3A2), are important for PCa growth. As illustrated in the model presented in *Figure 5*, GCN2 plays a critical role ensuring that appropriate expression of key *SLC* genes and associated amino acid transport are tightly linked to the ISR.

## GCN2 supports PCa tumor growth and maintains expression of amino acid transporters in mouse xenograft models

To examine the contributions of the GCN2/SLC transporter model for PCa growth in an in vivo model, we first selected the 22Rv1 cell line as it is castration resistant. Parental (WT) 22Rv1 cells with functional GCN2 or those with the eIF2α kinase deleted (GCN2 KO) were injected subcutaneously into the flanks of 8-week-old male NSG mice, and tumor volumes were measured for up to 12 days. Whereas the parental cells showed sustained tumor growth, loss of GCN2 significantly reduced tumor growth over time, resulting in a significant reduction in tumor weight at the end of the study (*Figure 6A*). IHC staining of tumor preparations showed diminished expression of Ki-67, a marker for cell proliferation in 22Rv1 tumors devoid of GCN2 (*Figure 6—figure supplement 1A*). As expected, the GCN2 protein was not detectable in the GCN2 KO tumors as judged by immunoblot, p-GCN2-T899 staining was reduced, and expression levels of the GCN2-targeted transporters 4F2 (SLC3A2), xCT (SLC7A11), and LAT1 (SLC7A5) were sharply reduced (*Figure 6B* and *Figure 6—figure supplement 1A*). Furthermore, the 22Rv1 tumors maintained expression of the ARv7 variant, which provides for castration resistance and resistance to enzalutamide (*Figure 6B*). Similar results were obtained in the AR-null castration-resistant cell line PC-3 following deletion of GCN2 (*Figure 6C, D*, *Figure 6—figure supplement 1B*,

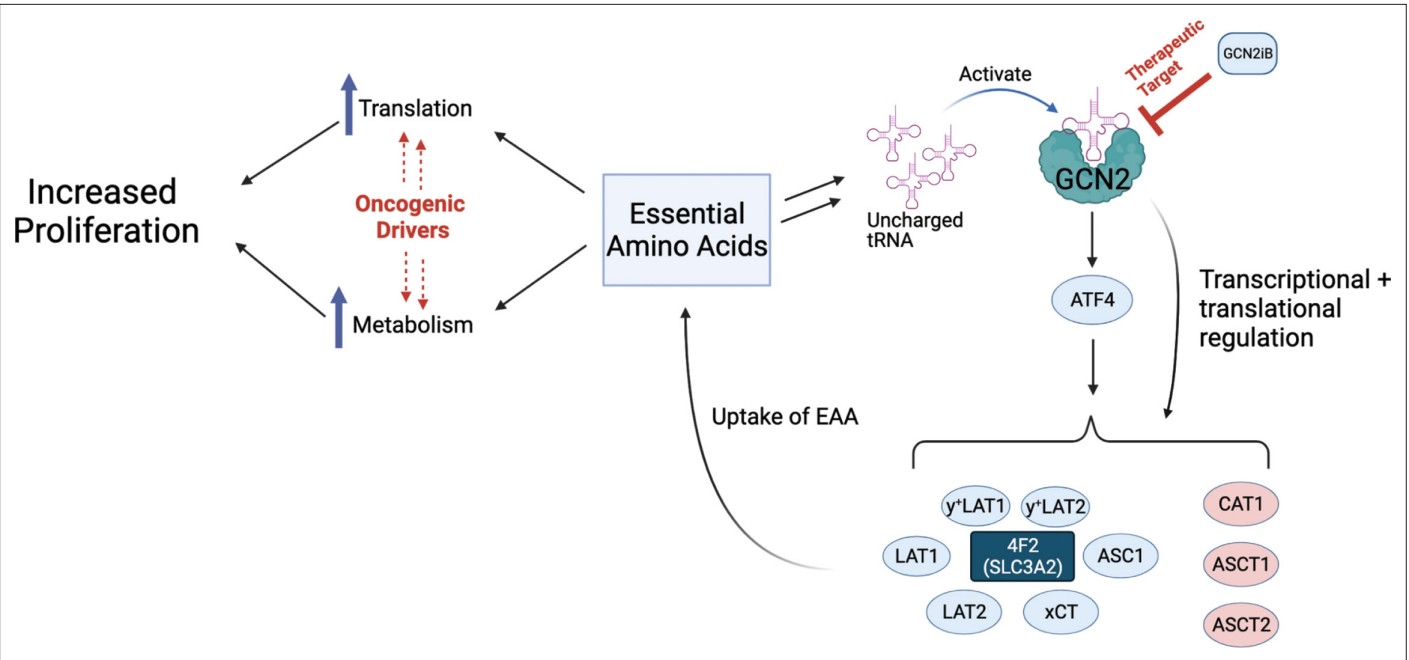

**Figure 5.** Model for GCN2 regulation of amino acid homeostasis in prostate cancer. Model depicting the role of GCN2 in regulating *SLC* amino acid transporters. Enhanced translation and altered metabolism driven by oncogenes deplete amino acid pools resulting in accumulation of uncharged tRNAs, leading to activation of GCN2. Active GCN2 results in increased expression of SLC amino acid transporters, including 4F2 (SLC3A2), to increase uptake of amino acids. Loss of GCN2 function disrupts amino acid homeostasis decreasing proliferation of prostate cancer cells.

and *Figure 6—figure supplement 1C*). Together, these xenograft experiments using the 22Rv1 and PC-3 models support the hypothesis that GCN2 provides significant growth advantages to PCa and suggests a mechanism involving enhanced SLC transporter expression that is independent of AR expression.

To determine the importance of ATF4 in tumor growth in vivo, we carried out a similar xenograft study using 22Rv1 cells deleted for ATF4. While loss of GCN2 led to dramatically reduced tumor growth, deletion of ATF4 only delayed tumor growth relative to the wild-type parental cells (*Figure 6E*). Furthermore, deletion of GCN2 led to a further decrease in 4F2 (SLC3A2), ASCT2 (SLC1A5), and CAT1 (SLC7A1) protein levels as compared to loss of ATF4 (*Figure 6—figure supplement 2A*). It is noted that loss of GCN2 did not reduce p-eIF2α levels in these end stage tumors, suggesting activation of a secondary eIF2 kinase or reduced phosphatase-directed dephosphorylation of eIF2α, which has been reported earlier in cells during extended nutrient limitation or other stress conditions (*Jiang et al., 2004*; *Zhang et al., 2002*). Interestingly, deletion of ATF4 resulted in increased activation of GCN2, increased levels of p-eIF2α, and in some cases increased amino acids relative to GCN2 KO tumors (*Figure 6F* and *Figure 6—figure supplement 2A*), indicating that loss of ATF4 can exacerbate cell stress facilitating further activation of GCN2. Consistent with these in vivo findings, knockdown of GCN2 expression in LNCaP cells in culture reduced amino acid levels to a greater degree as compared to knockdown of ATF4 (*Figure 6—figure supplement 2B*).

Another key feature of our GCN2/SLC transporter model is that this ISR regulator functions to ensure appropriate uptake of select EAA. To more directly test this model in vivo, we again utilized the xenograft model featuring 22Rv1 GCN2 KO cells. In this experiment, we provided additional EAA in the drinking water for the tumor bearing mice. While 22Rv1 cells with functional GCN2 showed tumor growth within 20 days of injection, loss of GCN2 reduced tumor growth even over the course of 40 days. However, consumption of EAA-supplemented water by GCN2 KO-tumor bearing mice provided for increased tumor growth approaching that of wild-type GCN2 tumors, albeit there was a time delay of 20 days for 22Rv1 with wild-type GCN2 to about 40 days with EAA-supplemented GCN2 KO (*Figure 6G*). Supplementation of the EAA did not affect tumor growth of parental 22Rv1 with functional GCN2 (*Figure 6G*). Similar to previous reports, supplementation of EAA in the drinking water resulted in a slight decrease in body weight of mice bearing either parental or GCN2 KO

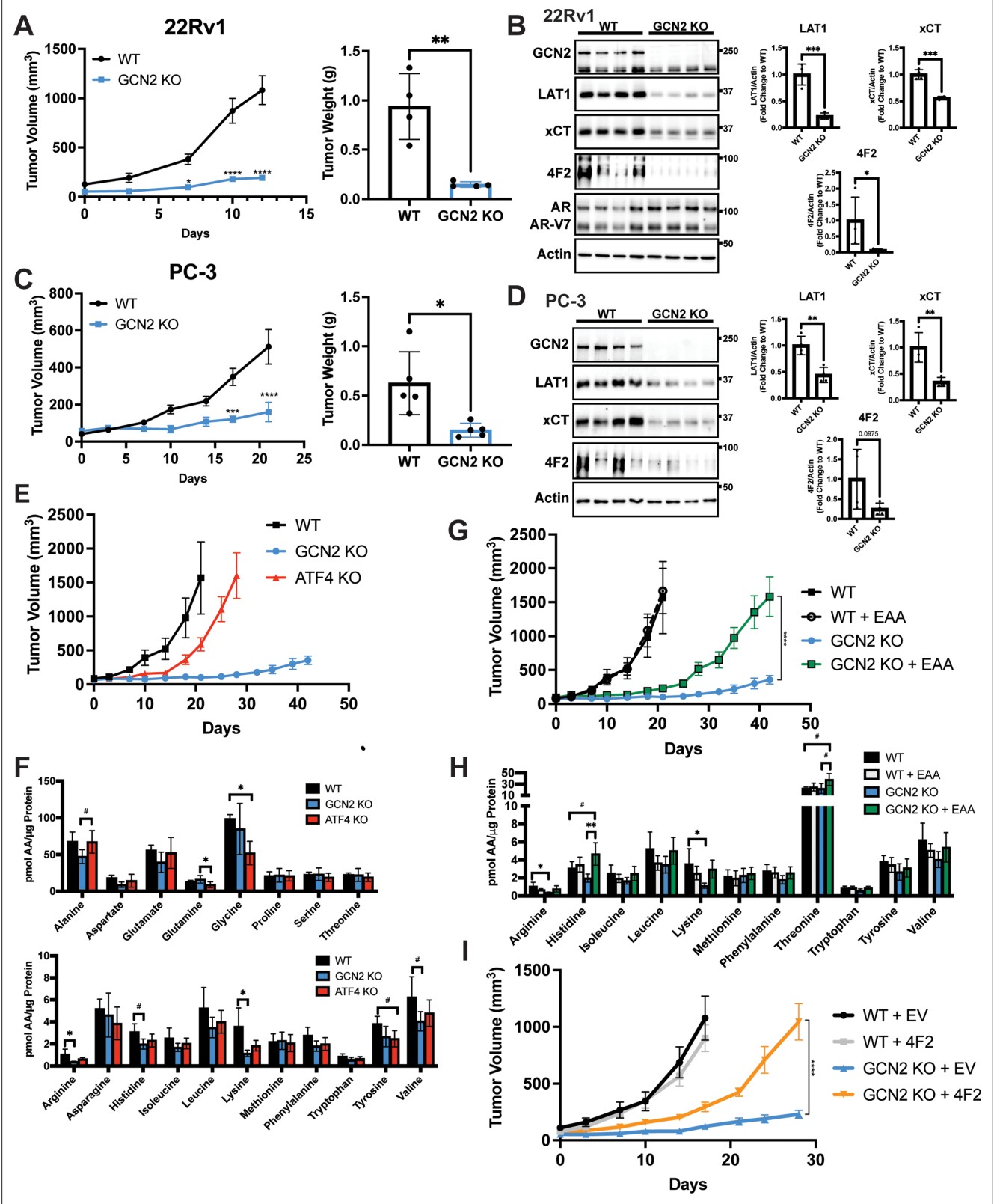

**Figure 6.** GCN2 regulates amino acid transporters ensuring sufficient amino acids for tumor growth in mouse prostate cancer (PCa) xenograft models. (**A**) WT or GCN2 KO 22Rv1 (clone 7) cells were injected subcutaneously into the dorsal flank of mice as described in the Materials and methods. Tumor volume (TV) was measured on indicated days and is plotted as average TV ± standard error of the mean (SEM) ($N = 4$). Statistical significance was determined using a two-way analysis of variance (ANOVA) with Sidak's multiple comparison; *$p \leq 0.05$; ****$p \leq 0.0001$. On the right bar graph, the final

*Figure 6 continued on next page*

*Figure 6 continued*

tumor weight was measured at endpoint and statistical significance was determined using an unpaired two-tailed *t*-test. Error bars indicate standard deviation (SD); **p ≤ 0.01. (**B**) Protein lysates were prepared from WT and GCN2 KO 22Rv1 tumors and analyzed by immunoblot to measure total GCN2, ATF4, LAT1 (SLC7A5), xCT (SLC7A11), 4F2 (SLC3A2), androgen receptor (AR), AR splice variant 7 (AR-V7), or actin. Molecular weight markers are indicated in kilodaltons for each immunoblot panel. The levels of the SLC proteins normalized to actin are shown in the bar graph (right panels). Statistical significance was determined using an unpaired two-tailed *t*-test. Error bars indicate SD (*N* = 4); *p ≤ 0.05; ***p ≤ 0.001. (**C**) Tumor growth of PC-3 WT and PC-3 GCN2 KO (clone 3) cells was analyzed in a mouse xenograft study as in A. Statistical significance was determined using a two-way ANOVA with Sidak's multiple comparison. Error bars indicate SEM (*N* = 5); ***p ≤ 0.001; ****p ≤ 0.0001. On the right bar graph, the final tumor weight was measured at endpoint. Statistical significance was determined using an unpaired two-tailed *t*-test. Error bars indicate SD (*N* = 5); *p ≤ 0.01. (**D**) Protein lysates were prepared from the PC-3 WT and PC-3 GCN2 KO tumors and analyzed by immunoblot for the indicated proteins. (Right panels) Quantification of protein levels of LAT1 (SLC7A5), xCT (SLC7A11), and 4F2 (SLC3A2) normalized to actin are shown in the bar graphs. Statistical significance was determined using an unpaired two-tailed *t*-test. Error bars indicate SD (*N* = 4); **p ≤ 0.01. (**E**) 22Rv1 WT (*N* = 4), 22Rv1 GCN2 KO (clone 7, *N* = 5), and 22Rv1 ATF4 KO (*N* = 5) were evaluated in the mouse xenograft model. Tumor volumes were measured on the indicated days. Error bars indicated SEM. (**F**) Amino acid measurements of 22Rv1 WT, 22Rv1 GCN2 KO, and 22Rv1 ATF4 KO tumors. Two separate bar graphs show high abundance (top) and low abundance (bottom) amino acids. Statistical significance was determined one-way ANOVA with Tukey's multiple comparisons. Error bars indicate SD (*N* = 4); #p ≤ 0.1, *p ≤ 0.05. (**G**) 22Rv1 WT and 22Rv1 GCN2 KO (clone 7) cells were analyzed in a xenograft model as described for (**A**), with or without supplementation of essential amino acid (EAA) in the drinking water. Tumor volume was measured on indicated days. 22Rv1 WT (*N* = 4) and 22Rv1 KO (*N* = 5) are the same tumor growth curves shown in (**E**). 22Rv1 WT + EAA (*N* = 5), 22Rv1 GCN2 KO + EAA (*N* = 5). Statistical significance was determined using a two-way ANOVA with Sidak's multiple comparison; ****p ≤ 0.0001. (**H**) Amino acid measurements for 22Rv1 WT, 22Rv1 GCN2, and 22Rv1 GCN2 KO + EAA tumors. Bar graphs show only amino acids present in EAA supplemented water. Statistical significance was determined using a one-way ANOVA with Tukey's multiple comparisons. Error bars indicated SD (*N* = 4); #p ≤ 0.1; *p ≤ 0.05; **p ≤ 0.01. (**I**) Tumor growth curves for 22Rv1 WT or 22Rv1 GCN2 KO (clone 11) transduced with 4F2 (SLC3A2) lentivirus (WT + 4F2 and GCN2 KO + 4F2) or empty vector (WT + EV and GCN2 KO + EV). Tumor volumes were measured on indicated days. Error bars indicate SEM (*N* = 5). Statistical significance was determined using a two-way ANOVA with Sidak's multiple comparison; ****p ≤ 0.0001.

The online version of this article includes the following figure supplement(s) for figure 6:

**Figure supplement 1.** Deletion of GCN2 reduces tumor growth and proliferation in 22Rv1 and PC-3 tumors.

**Figure supplement 2.** Deletion of GCN2 or ATF4 reduces the expression of SLC genes associated with amino acid transport in 22Rv1 tumors.

**Figure supplement 3.** Effect of essential amino acid (EAA) supplementation on mouse weight, amino acid levels, and expression of SLC genes associated with amino acid transport in 22Rv1 WT and GCN2 KO tumors.

**Figure supplement 4.** Effect of overexpression of 4F2 (SLC3A2) on amino acid levels in 22Rv1 WT and GCN2 KO tumors.

tumors (*Corsetti et al., 2014*; *Figure 6—figure supplement 3A*). Measurements of free amino acids in tumors indicated that loss of GCN2 reduced intracellular arginine and lysine concentrations, with histidine and the branched-chain amino acids trending lower. Dietary supplementation of EAA partially restored these amino acid levels, although only histidine achieved statistical significance (*Figure 6H* and *Figure 6—figure supplement 3B*). Immunoblot measurements showed that loss of GCN2 lowered key ISR markers, including the key SLC transporters independent of the supplementation of EAA (*Figure 6—figure supplement 3C*). We also tested the impact of overexpression of 4F2 (SLC3A2) on growth of 22Rv1 tumors. Similar to supplementation with EAA, overexpression of 4F2 (SLC3A2) in GCN2 KO tumors partially rescued growth and restored tumor amino acid levels (*Figure 6I*, *Figure 6—figure supplement 4A*, and *Figure 6—figure supplement 4B*). These results further support the notion that GCN2 has a primary function in maintaining EAA and highlight a key role for GCN2 in the regulation of 4F2 (SLC3A2) expression. In keeping with the cultured PCa cell line experiments, GCN2 played a vital role supporting amino acid homeostasis in these tumor models.

## Pharmacological inhibition of GCN2 reduces tumor growth in xenograft models

We next tested the impact of pharmacological inhibition of GCN2 function in cancer cell-derived and patient-derived xenograft models. Male NSG mice were injected subcutaneously with LNCaP or 22Rv1 cells or implanted with tumor fragments from an androgen-sensitive tumor TM00298 (*Sekhar et al., 2019*). Alternatively, male castrated NSG mice were implanted with tumor fragments from LuCaP-35 castration-resistant tumors (*Corey et al., 2003*). Tumors were allowed to grow to 50–100 mm³ and mice were randomized to vehicle or GCN2iB treatment groups. Treatment with GCN2iB at 30 mg/kg twice daily for 2–6 weeks resulted in robust anti-tumor activity in all four models as shown by a significant delay in the tumor volume, reductions in tumor mass at endpoint, and reduced expression of p-GCN2-T899 and Ki-67 (*Figure 7A–E* and *Figure 7—figure supplement 1A*).

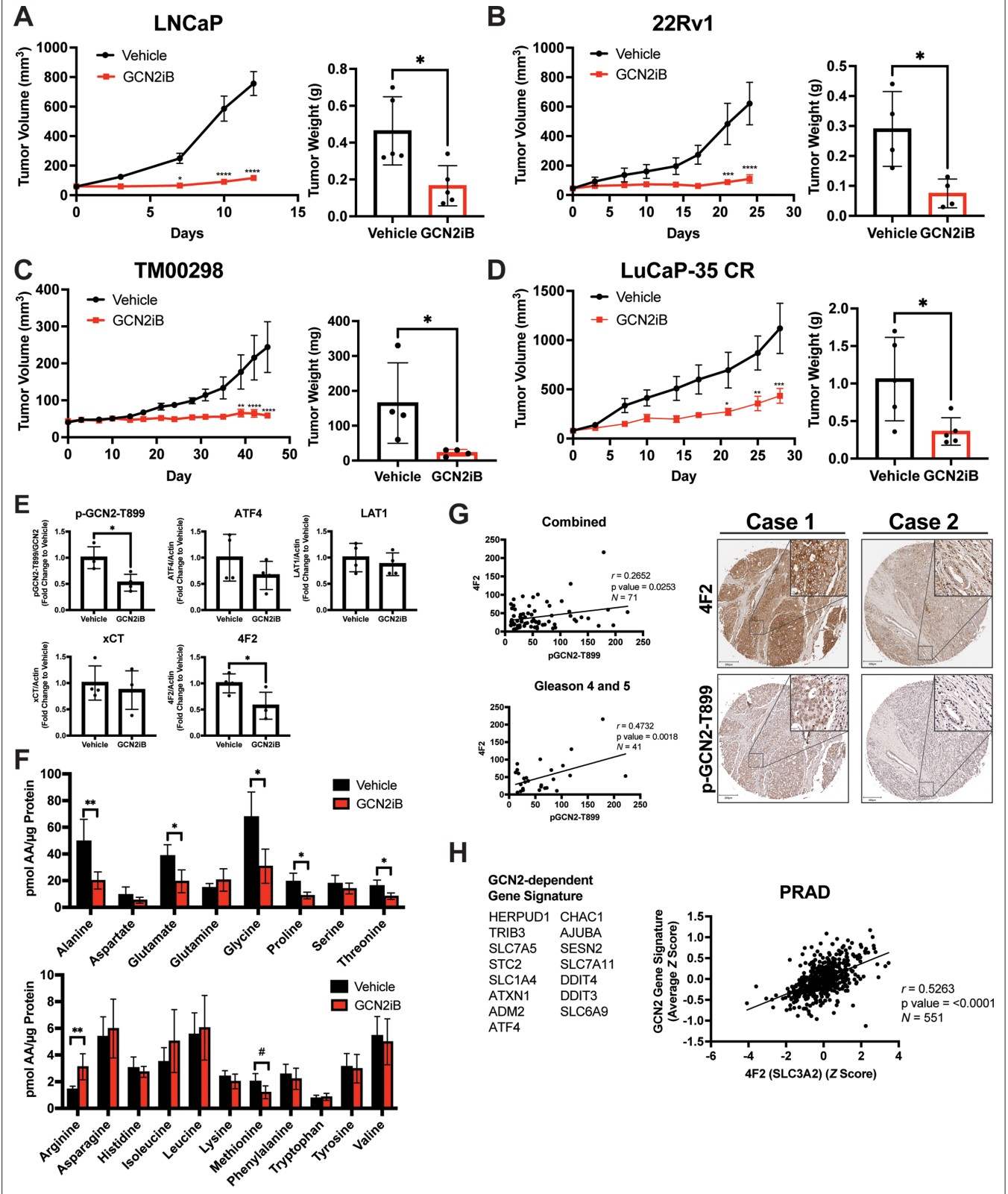

**Figure 7.** Pharmacological inhibition of GCN2 reduces tumor growth in cell line-derived and patient-derived xenograft models. Male NSG mice were injected subcutaneously with LNCaP (N = 5) (**A**) or 22Rv1 (N = 4) (**B**) cells, or alternatively implanted with tumor fragments from an androgen-sensitive tumor TM00298 (N = 5) (**C**). Male castrated NSG mice were implanted with tumor fragments from LuCaP-35 CR tumors (N = 5) (**D**). Mice were treated with vehicle or 30 mg/kg GCN2iB twice daily for 5 days/week and tumor volumes were measured on indicated days. Statistical significance

*Figure 7 continued on next page*

*Figure 7 continued*

was determined using a two-way analysis of variance (ANOVA) with Sidak's multiple comparison. Error bars indicate standard error of the mean (SEM); *p≤0.05, **p≤0.01, ***p ≤ 0.001, ****p≤0.0001. Final tumor weight was measured at endpoint and is represented in bar graphs (right panels). Statistical significance was determined using an unpaired two-tailed *t*-test. Error bars indicate standard deviation (SD); *p ≤ 0.05. (**E**) Protein lysates were prepared from 22Rv1 tumors treated with vehicle or GCN2iB and analyzed by immunoblot for phosphorylated GCN2-T899, ATF4, LAT1 (SLC7A5), xCT (SLC7A11), and 4F2 (SLC3A2), and actin. The levels of the SLC proteins normalized to actin are shown. Phosphorylated GCN2-T899 was normalized to total GCN2. Statistical significance was determined using an unpaired two-tailed *t*-test. Error bars indicate SD (*N* = 4); *p≤0.05. (**F**) Amino acid measurements of 22Rv1 tumors treated with vehicle or GCN2iB. Bar graphs show high abundance (top) and low abundance (bottom) amino acids. Statistical significance was determined using an unpaired two-tailed *t*-test. Error bars indicate SD (*N* = 4); #p≤0.1; *p≤0.05; **p≤0.01. (**G**) Pearson correlation between p-GCN2-T899 and 4F2 (SLC3A2) histoscores calculated from IHC staining from a prostate tumor microarray (Biomax PR807c) containing normal (*N* = 10), hyperplasia (*N* = 20), and malignant (*N* = 50) for all tissues (combined) or Gleason scores 4 and 5. The center lines depict linear regression (95% confidence intervals). Not all samples were analyzed due to damaged/quality of tissue samples. Levels of p-GCN2-T899 and 4F2 (SLC3A2) were measured by IHC staining and QuPath was used to determine the histoscore. Two representative cases are shown for high (Case 1) and low (Case 2) p-GCN2-T899 and 4F2 (SLC3A2) staining. Scale bar indicates 200 µm (main image) and 20 µm (insert). (**H**) Correlation of expression of 4F2 (SLC3A2) and a GCN2-dependent gene signature in prostate adenocarcinoma (PRAD, *N* = 551) from the Cancer Genome Atlas (TCGA). The GCN2-dependepent gene signature was derived from RNA-seq data as described in the Materials and methods.

The online version of this article includes the following figure supplement(s) for figure 7:

**Figure supplement 1.** GCN2iB reduces proliferation in prostate cancer (PCa) cell line-derived and patient-derived xenograft models and has no effect on mouse body weight.

**Figure supplement 2.** Treatment with GCN2 inhibitor, GCN2iB, reduces the expression of SLC genes associated with amino acid transport in 22Rv1 and TM00298 tumors.

**Figure supplement 3.** 4F2 (SLC3A2) protein levels are higher in malignant prostate cancer tissue compared to normal prostate tissue.

Importantly, GCN2iB had no effect on the body weight of treated mice over the course of treatment (*Figure 7—figure supplement 1B*), lowered expression of 4F2 (SLC3A2) in both 22Rv1 and TM00298 tumors, and reduced levels of select amino acids in 22Rv1 tumors from treated mice (*Figure 7E, F, Figure 7—figure supplement 2A*, and *Figure 7—figure supplement 2B*). These results suggest that GCN2 is important for tumor growth and expression of 4F2 (SLC3A2) in both androgen-sensitive and castration-resistant tumors.

Our results suggest that GCN2 controls expression of multiple amino acid transporters to regulate amino acid homeostasis in PCa tumors. We next addressed whether expression of p-GCN2- or GCN2-dependent mRNA transcripts correlated with expression of 4F2 (SLC3A2) in human tumor samples. Again, using a tumor tissue array containing patient samples, we stained for p-GCN2-T899 and 4F2 (SLC3A2). Similar to recent findings (*Maimaiti et al., 2021*), 4F2 (SLC3A2) staining was increased in malignant tissue samples as compared to normal tissue (*Figure 7—figure supplement 3A*). Interestingly, SLC3A2 (4F2) showed a moderate, albeit significant, correlation with expression of p-GCN2-T899 (*r* = 0.27, p = 0.025). This correlation was more striking among tissues from higher grade tumors (*r* = 0.47, p = 0.002, Gleason scores 4 and 5) (*Figure 7G*), which was further supported by staining of p-GCN2 and 4F2 (SLC3A2) of core needle biopsy specimens from patients with high-grade PCa (*Figure 7—figure supplement 3B*). To further test the GCN2/SLC3A2 (4F2) model, we derived a gene signature from GCN2-dependent transcripts from our RNA-seq analysis and measured a significant correlation with SLC3A2 (4F2) expression (*r* = 0.53, p < 0.0001) using a cohort of prostate cancer patients (PRAD) from the TCGA database (*Figure 7H*). These results further support the idea that pharmacological inhibition of GCN2 may serve as a potent strategy for the treatment of androgen-sensitive and castration-resistant PCa by limiting the expression of amino acid transporters including 4F2 (SLC3A2).

## Discussion

Tumor cells often demonstrate high demand for amino acids to sustain their increased metabolism, protein synthesis, and proliferation (*Lieu et al., 2020*), and frequently overexpress amino acid transporters to meet this increased demand (*Bröer, 2020*; *Bhutia et al., 2015*). Altered or increased transport of EAA and NEAA has been suggested to be a point of vulnerability in several cancers (*Pathria and Ronai, 2021*; *Butler et al., 2021*). In this study, we show that the adaptive ISR pathway driven by GCN2 is active in PCa tissue and can be genetically and pharmacologically targeted in human models of androgen-sensitive and castration-resistance disease. Loss of GCN2 function by genetic deletion

or pharmacological inhibition sharply reduced PCa growth in vitro and reduces tumor growth in xenograft models. As illustrated in *Figure 5*, GCN2 directs the expression of a large number of SLC transporter genes, many of which are critical for PCa growth. Among these GCN2-targeted genes are 4F2 (SLC3A2), a type II integral membrane protein which is known to form dimers with SLC7 family members including SLC7A5 (LAT1) and SLC7A11 (xCT), each contributing to uptake of specific amino acids (*Pizzagalli et al., 2021*). Loss of GCN2 in PCa reduced free amino acids both in vitro and in tumors derived from xenograft models. Addition of EAAs to the medium of cultured cells rescued the growth defect associated with GCN2 inhibition; furthermore, supplementation of EAAs in the drinking water of mice injected subcutaneously with GCN2 KO PCa cells substantially enhanced tumor growth in vivo. While several strategies to deprive cancer cells of nutrients have been contemplated (*Pathria and Ronai, 2021*), our results suggest that inhibition of GCN2 represents a unique treatment paradigm to starve prostate tumors of EAAs. Further supporting the utility of GCN2 as a therapeutic target for PCa, a small molecule inhibitor of GCN2 significantly slowed tumor growth in both CDX and PDX models of androgen-sensitive and castration-resistant disease irrespective of AR status.

GCN2 is activated by uncharged tRNAs that accumulates during amino acid starvation or by ribosome collisions (*Wek et al., 1995*; *Wu et al., 2020*; *Zaborske et al., 2009*; *Wek et al., 1989*). As indicated by p-GCN2, this eIF2α kinase is induced in a variety of different proliferating PCa cell lines and in malignant prostate tissue. While our results suggest that amino acid limitations and accompanying uncharged tRNA contribute to GCN2 activation in PCa, it is noted that other mechanisms can contribute to regulation of GCN2. For example, a mitochondrial Rho GTPase 2 (MIRO2) overexpressed in PCa was recently shown to bind to GCN1, a critical activator of GCN2 (*Furnish et al., 2022*), and certain mitochondrial defects were reported to contribute to activation of GCN2 and the ISR (*Mick et al., 2020*). Given that these different PCa models represent those that are androgen-sensitive, castration-resistant, resistant to enzalutamide, or lack AR expression altogether, our study suggests that GCN2 functions to control PCa growth independent of AR function. Supplementation of EAAs reduces p-GCN2 and partially rescued the growth deficiency of GCN2 KO cells, suggesting that deficiency of at least some of these critical amino acids are limiting in proliferating PCa cells. This nutrient limitation would trigger GCN2 induction of the ISR which features induced expression of SLC transporters, including the central 4F2 (SLC3A2), and these events are required to sustain tumor growth. It is noteworthy that 4F2 (SLC3A2) has recently been shown to be overexpressed in PCa and correlated with reduced progression-free survival (*Maimaiti et al., 2021*).

Experiments involving supplementation of individual amino acids to cultured LNCaP cells knocked down for GCN2 suggest that this eIF2α kinase is critical for maintenance of histidine homeostasis. Histidine is an EAA whose transport is not clearly understood; although, LAT1 (SLC7A5) has been suggested to facilitate its uptake (*Scalise et al., 2018*). Beyond its critical role in protein synthesis, histidine also functions as an important antioxidant (*Son et al., 2005*) and its catabolism has been linked to nucleotide synthesis in some cancers (*Kanarek et al., 2018*). Additional studies will be required to delineate how histidine contributes to PCa metabolism and growth. Whereas histidine also rescued the growth defect of GCN2 inhibition in the related MR49F cells, only lysine alleviated the slow growth of cultured 22Rv1 GCN2 KO cells. These in vitro experiments suggest a more complex picture where the limiting amino acid is cell-type specific or influenced by specific genetic alterations which might influence the threshold or mechanism of GCN2 activation. It is acknowledged that amino acids included in the media used in the in vitro experiments are typically present at high physiological levels so the importance of any given EAA may be masked by the abundance of these nutrients in the culture medium. Consistent with this notion, the analysis of 22Rv1 GCN2 KO tumors showed multiple EAAs were reduced.

Activation of GCN2 triggers the ISR by the induction of ATF4, a key transcriptional activator. ATF4 contributes to PCa growth induced by GCN2, as depletion of ATF4 in multiple PCa cell lines led to growth inhibition in culture. Previous studies have suggested that ATF4 is important for growth in multiple cancers (*Tameire et al., 2019*; *Wortel et al., 2017*) including PCa (*Pällmann et al., 2019*). However, under our experimental conditions, whereas loss of GCN2 led to a sharp reduction in tumor growth in the 22Rv1 xenograft model, deletion of ATF4 only delayed tumor growth. These results suggest that while ATF4-directed transcriptional activation is important for PCa proliferation, GCN2 functions by additional mechanisms, including preferential translation of multiple ISR-target genes.

As noted above activated GCN2 induces a program of genes that are important for maintenance of amino acid homeostasis. It should also be noted that another major nutrient sensing pathway is regulated by mTORC1 (*Liu and Sabatini, 2020*) and mTORC1 activity is transiently reduced in GCN2iB-treated cells (data not shown). We focused on the critical role of GCN2 in controlling expression of 4F2 (SLC3A2), albeit a collection of *SLC* genes are required GCN2 for full expression. While 4F2 (SLC3A2) is an important contributor to amino acid homeostasis, deletion of 4F2 (SLC3A2) in LNCaP cells led to depletion of only a portion of the free amino acids that were disrupted by loss of GCN2, consistent with its requirement with a subset of amino acid transporters (*Pizzagalli et al., 2021*). Furthermore, overexpression of 4F2 (SLC3A2) only partially restored growth of 22Rv1 GCN2 KO tumors in vivo and did not affect the progression of parental 22Rv1 tumors. These results indicate that the central role of GCN2 for ensuring appropriate amino acid homeostasis and enabling PCa proliferation involves induction of multiple SLC transporters. In addition, the reduction in ATF4 expression that accompanies loss of GCN2 function reduces expression of several amino acid biosynthetic genes which would exacerbate the observed effects on uptake of NEAAs. Taken together, these observations suggest that inhibition of GCN2 may be a more effective therapeutic strategy for starving PCa of amino acids as compared to therapies that target individual SLC transporters. In summary, our results indicate that GCN2 may represent a novel therapeutic target for PCa. The therapeutic landscape for PCa patients is rapidly evolving but AR still remains the focus. Unfortunately, resistance to androgen deprivation therapies continue to be a major challenge. The development of new therapeutic strategies that work independently from inhibition of the AR axis is highly clinically relevant and based on our findings, inhibitors of GCN2 should be considered further as part of a strategy for treatment of androgen-sensitive and castration-resistant PCa.

## Materials and methods
### Cell culture and treatments

LNCaP clone FGC (Cat. #CRL-1740, RRID:CVCL_1379), C4-2B (Cat. #CRL-3315, RRID:CVCL_4784), CW22Rv1 (referred to as 22Rv1, Cat. #CRL-2505, RRID:CVCL_1045), PC-3 (Cat. #CRL-1435, RRID: CVCL_0035), and HEK-293T (Cat. #CRL-3216, RRID:CVCL_0063) cells were purchased from the American Tissue Type Collection (ATCC, Manassas, VA). BPH-1 (Cat. #SCC256, RRID: CVCL_1091) cells were purchased from MilliporeSigma (Temecula, CA). LAPC-4 (RRID:CVCL_4744) cells were provided by Dr. Dominic J. Smiraglia (Roswell Park Comprehensive Cancer Center, Buffalo, NY). MR49F (RRID:CVCL_RW53) cells were obtained from Dr. Timothy Ratliff (Purdue University, West Lafayette, IN). LNCaP, C4-2B, 22Rv1, and BPH-1 cells were maintained in phenol-free RPMI-1640 (Gibco, Cat. #11835-030) supplemented with 10% fetal bovine serum (FBS) (HyClone, Cat. #SH30071). LAPC-4 cells were maintained in phenol-free RPMI-1640 supplemented with 10% FBS as described above and 10 nM dihydrotestosterone (Sigma-Aldrich, Cat. #D-073). MR49F cells were maintained in phenol-free RPMI-1640 supplemented with 10% FBS as described above and 10 μM enzalutamide (MedChemExpress, Cat. #HY-70002). PC-3 cells were maintained in Ham's F-12K (Kaighn's) media (Gibco, Cat. #21127-022) or Human Plasma-Like Media (HPLM, Gibco Cat. #A4899101) supplemented with 10% FBS. HEK-293T cells were maintained in DMEM (Corning, Cat. #10-013-CV) supplemented with 10% FBS. All cell lines were maintained at 37°C in a fully humidified atmosphere containing 5% $CO_2$, used at early passage for experiments, and tested to be free of mycoplasma contamination. For cell treatments, cells were plated at 250,000 cells per well in 6-well plates in standard culture media as described above and allowed to attach overnight. Media was changed to standard culture media containing 500 nM to 10 μM GCN2iB or DMSO as a control or standard media lacking histidine (MyBiosource, Cat. #MBS652918) and incubated for the indicated time period. GCN2iB (*Nakamura et al., 2018*) was synthesized at Sun-shinechem (Wuhan, China). As indicated, salubrinal (Tocris Biosciences, Cat. #405060-95-9) was added to the culture media at a concentration of 50 μM. To measure bulk protein translation, 1 μM puromycin (MP Biomedicals Cat. #194539) was included in the culture medium for the last 15 min of treatment with GCN2iB, and protein lysates were probed and quantified by immunoblot analyses using anti-puromycin antibody (*Schmidt et al., 2009*).

## Immunoblot analysis

Whole cell extracts and tumor lysates were prepared in 1% sodium dodecyl sulfate (SDS) lysis buffer supplemented with 1× Halt Protease and Phosphatase Inhibitor Cocktail (Thermo Scientific, Cat. #1861281). Tumors were homogenized using a Kinematica Polytron tissue homogenizer. Lysates were boiled for 10 min, sonicated, and subjected to centrifugation at 20,000 × $g$ to remove insoluble material. Protein concentrations were determined by DC Protein Assay (Bio-Rad, Cat. #5000112) in technical triplicate using bovine serum albumin as the protein standard. Equal amounts of protein samples, which ranged from 10 to 20 µg between experiments, were separated on SDS–polyacrylamide gel electrophoresis gels, transferred to nitrocellulose membranes, and used for immunoblot analysis. The primary antibodies used were as follows: phospho-GCN2-T899 (Abcam Cat. #ab75836, RRID:AB_1310260), total GCN2 (Cell Signaling Technology Cat. #3302, RRID:AB_2277617), total PERK (Cell Signaling Technology Cat. #3192, RRID:AB_2095847), total HRI (Santa Cruz Biotechnology Cat. #sc-365239, RRID:AB_10843794), total PKR (Cell Signaling Technology Cat. #12297, RRID:AB_2665515), phospho-eIF2α-S51 (Abcam Cat. #ab32157, RRID:AB_732117), total eIF2α (Cell Signaling Technology Cat. #5324, RRID:AB_10692650), ATF4 (Cell Signaling Technology Cat. #11815, RRID:AB_2616025), or custom rabbit polyclonal antibody which was prepared against full-length recombinant human ATF4 protein and affinity purified, ASNS (Cell Signaling Technology, Cat. #20843S), TRIB3 (Abcam Cat. #ab75846, RRID:AB_1310768), GADD34 (Proteintech Cat. #10449-1-AP, RRID:AB_2168724), SLC7A5/LAT1 (Cell Signaling Technology Cat. #5347, RRID:AB_10695104), SLC7A11/xCT (Cell Signaling Technology Cat. #12691, RRID:AB_2687474), SLC3A2/4F2 (Cell Signaling Technology Cat. #47213, RRID:AB_2799323), SCL7A1/CAT1 (Proteintech Cat. #14195-1-AP, RRID:AB_2190723), SLC1A4/ASCT1 (Cell Signaling Technology Cat. #8442, RRID:AB_10828382), SLC1A5/ASCT2 (Cell Signaling Technology Cat. #5345, RRID:AB_10621427), AR (Cell Signaling Technology Cat. #5153, RRID:AB_10691711), puromycin (Millipore Cat. #MABE343, RRID:AB_2566826), β-actin (Sigma-Aldrich Cat. #A5441, RRID:AB_476744), and β-tubulin (Cell Signaling Technology Cat. #2146, RRID:AB_2210545). Immunoblot signals were visualized by enhanced chemiluminescence (ECL) using Clarity Western ECL Substrate (Bio-Rad, Cat. #170-5060) or SuperSignal West Femto Maximum Sensitivity Substrate (Thermo Fisher Scientific, Cat. #34094) and images were captured using a ChemiDoc Imaging System (Bio-Rad, RRID:SCR_019037). Prestained protein standards (Bio-Rad Cat. #1610377) were included on each immunoblot, captured as a separate merged image file, and the molecular weights are indicated in each immunoblot panel.

## RNA interference and proliferation assays

Small interfering RNAs (siRNAs) were purchased from Dharmacon and sequences are shown in *Supplementary file 5*. The siRNAs that showed the most robust depletion of the targeted genes in the indicated cells were selected for experiments. For growth assays LAPC-4, C4-2B, MR49F, 22Rv1, or PC-3 cells were seeded into 96-well plates at 2500–5000 cells per well in normal growth media as described above and allowed to attach overnight. Cells were transfected with 1 pmol of siRNA using Lipofectamine RNAiMAX Transfection Reagent (Thermo Scientific, Cat. #13778030) or DharmaFECT 3 Transfection Reagent (Horizon, Cat. #T-2003-01) according to the manufacturer's protocol and incubated overnight (considered day 0). Alternatively, where indicated for LAPC-4 cells, growth media was changed to serum-free Accell Delivery Media (Horizon, Cat. #B-005000) and transfected with 1 pmol of Accell siRNA and incubated 48 hr at which time cells were changed back to normal growth media (considered day 0). Cells were incubated for an additional 2–6 days and cell growth was quantified using CellTiter-Glo 2.0 Reagent (Promega, Cat. #924C). For immunoblot analysis, PCa cells were seeded into 6-well plates at 250,000–350,000 cells per well and transfected with 30 pmol of siRNA as described above and incubated for 48 hr prior to lysate preparation as described above.

## In vitro amino acid supplementation

For amino acid supplementation experiments, 1× MEM Amino Acids Solution (Gibco, Cat. #11130051) was used for EAA mix and 1× MEM NEAA solution (Gibco, Cat. #11140050) was used for NEAA mix. Experiments utilizing single amino acids were performed using the following concentrations: 600 µM L-arginine hydrochloride, 100 µM L-cystine dihydrochloride, 200 µM L-histidine hydrochloride, 400 µM L-isoleucine, 400 µM L-leucine, 395 µM L-lysine hydrochloride, 100 µM L-methionine, 200 µM

L-phenylalanine, 400 µM L-threonine, 50 µM L-tryptophan, 200 µM L-tyrosine, and 400 µM L-valine. All amino acids were dissolved in water and were purchased from Sigma-Aldrich (St. Louis, MO).

## Amino acid uptake measurements

LNCaP, parental 22Rv1, or 22Rv1 GCN2 KO cells were plated in 24-well plates (50,000 cells) in standard culture media as described above. LNCaP cells were treated with vehicle (DMSO) or 2 µM GCN2iB for 24 hr. Cells were washed with pre-warmed sodium-free Hank's Balanced Salt Solution (HBSS) and treated with 100 nM $^3$H-arginine (5 µCi/ml, Perkin-Elmer), 1 µM $^{14}$C-leucine (0.3 µCi/ml, Perkin-Elmer), or 1 µM $^{14}$C-lysine (0.3 µCi/ml, Perkin-Elmer) in HBSS containing the appropriate unlabeled amino acid (100–200 µM arginine, 1–20 µM leucine, or 1 µM lysine) at 25°C for 2 min. Transport was stopped by adding the appropriate excess cold-amino acid (non-radioactive) in HBSS. Cells were washed with phosphate-buffered saline (PBS), lysed with Passive lysis buffer (Promega, Cat. #E1941), and a portion added to scintillation cocktail and counted. The protein concentration of each lysate was determined as described above and amino acid uptake was represented as nmol/µg of total protein/min. Each measurement was performed in triplicate.

## Plasmid construction

Plasmid pDONR221-SLC3A2_STOP was a gift from RESOLUTE Consortium & Giulio Superti-Furga (Addgene plasmid # 161379, RRID:Addgene_161379) (*Girardi et al., 2020*). Plasmid pENTR223-1 GCN2 (NM_001013703) was purchased from Transomic Technologies and pLX303 was a gift from David Root (Addgene plasmid # 25897, RRID:Addgene_25897). GCN2 coding sequence was subcloned into the lentiviral vector pLVX-IRES-Hygromycin (Clontech, Cat. #632185) using an In-Fusion HD Cloning plus kit (TaKaRa, Cat. #638920) according to the manufacturer's protocol yielding plasmids pLVX-GCN2-hygro. The SLC3A2 (4F2) coding region was subcloned into the lentiviral vector pLX303 by Gateway cloning using Gateway LR Clonase II Enzyme Mix (Invitrogen, Cat. #11791-020) according to the manufacturer's protocol yielding plasmid pLX303-4F2-blast. Plasmid pMSCV-GADD34-puro contains the human PPP1R15A (GADD34, NM_014330.5) gene subcloned into the NcoI/EcoRI sites of the retroviral vector pMSCV-puro (Clontech Laboratories, Mountain View, CA).

## Construction of cell lines

GCN2 and ATF4 knockout cell lines were generated using CRISPR/Cas9 Human Gene Knockout Kits (Origene, Cat. #KN412459 and #KN402333) using hGCN2g1 (5′-AATTTAGTTTTGTACCCTCA-3′) and hATF4g1 (5′-CTTCCTGAGCAGCGAGGTGT-3′), respectively, following the manufacturer's protocol. Cell lines overexpressing GCN2 or SLC3A2 (4F2) were created by lentivirus transduction. Briefly, HEK-293T cells were cultured in DMEM as described above and were transfected with lentivirus constructs and ViraPower Lentiviral Packaging Mix (Invitrogen, Cat. #35-1275) using Lipofectamine 2000 (Thermo Fisher Scientific, Cat. #11668030) according to the manufacturer's protocol and virus containing supernatant was collected over 48–72 hr, concentrated using Lenti-X Concentrator (TaKaRa, Cat. #631232), and stored in aliquots at −80°C. 22Rv1 or PC-3 cells were transduced with virus-containing culture media supplemented with 10 µg/ml Polybrene (Millipore, Cat. #TR-1003-G) for 24 hr and replaced with fresh growth media. At 72 hr post-transfection, growth media was changed to media supplemented with 250 µg/ml Hygromycin B (Thermo Cat. #10687010) or 10 µg/ml blastocidin S (Sigma, Cat. #SBR00022) to select for stable pools as appropriate. The appropriate empty vector was used as a control for all experiments.

## qRT-PCR analysis

Total RNA was isolated from cultured cells or tumor tissue using TRIzol LS Reagent (Invitrogen, Cat. #10296010) and quantified using a Nanodrop Spectrophotometer (Thermo Fisher Scientific). Total RNA (1 µg) was converted to cDNA using a High-Capacity cDNA Reverse Transcription Kit (Thermo Fisher Scientific, Cat. #4368813) according to the manufacturer's protocol. Following cDNA synthesis, samples were diluted fivefold in molecular biology grade water and PCR amplification was carried out using PowerUp SYBR Green Master Mix (Thermo Fisher Scientific, Cat. #A25742) using an Applied Biosystems QuantSudio5 PCR system. The primers used for measuring *ATF4* and *SLC3A2 (4F2)* transcript levels are shown in *Supplementary file 5*. The relative abundance of each transcript was

calculated using the ΔΔCT method with *GAPDH* serving as an internal control and data are presented normalized to each control group.

## Cell cycle analysis

Cell cycle analysis on LNCaP, MR49F, C4-2B, 22Rv1, and PC-3 cells was performed by costaining with a Click-iT EdU Alexa Fluor 488 Flow Cytometry Assay Kit (Invitrogen, Cat. #C10425) and FxCycle Violet Ready Flow Reagent (Invitrogen, Cat. #R37166) according to the manufacturer's protocol. Cell treatments and siRNA knockdowns were carried out for 48 hr as described above before analysis using an LSR II (LSR4) flow cytometer (BD Biosystems). A minimum of 10,000 events were analyzed per condition and all experiments were performed using triplicate wells. Data analysis was performed using FlowJo (FlowJo LLC, TreeStar) and data are presented as the percentage of cells in G1, S, or G2-M phases of the cell cycle.

## Amino acid analysis

Levels of different amino acids were measured in cultured cells or tumor lysates by HPLC or liquid chromatography–tandem mass spectrometry (LC–MS). Cell pellets or tissues were resuspended in 0.1% formic acid in methanol, vortexed briefly, and subjected to three cycles of freezing in liquid nitrogen followed by thawing at 42°C for 10 min. Clarified lysates were analyzed by reverse-phase LC with appropriate standard curves as described previously (*Nikonorova et al., 2018*) or using LC–MS/MS with stable isotope dilution. For the MS-based method, lysates were spiked with a mixture containing 37 amino acid isotopic internal standards and deproteinated using a solution of 90% acetonitrile, 10 mM ammonium formate, and 0.15% formic acid. Clarified lysates were analyzed using a biphasic LC–MS/MS approach first with a HILIC separation (*Prinsen et al., 2016*) followed by a mixed-mode chromatographic separation (*Griffin et al., 2019*). Quantification of amino acid levels was determined by fitting response ratios to an eight-point calibration curve generated using verified reference material for each of the 43 amino acids quantified.

## CRISPR knockout screen

A human SLC knockout library (*Girardi et al., 2020*) was purchased from Addgene, (Cat. #132552). Lentivirus was generated as described above. LNCaP and 22Rv1 cells were transduced at a multiplicity of infection of 0.5 and selected with 1 µg/ml puromycin (MP Biomedicals Cat. #194539) for 7 days. Three independent screens for both LNCaP and 22Rv1 were performed. Cells were then cultured for an additional 4 weeks and gDNA was isolated using DNeasy Blood and Tissue Kit (Qiagen, Cat. #69506). sgRNA sequences were PCR amplified from gDNA using Platinum SuperFi II DNA Polymerase (Thermo Fisher Scientific, Cat. #12369002) and primers sgRNA_F and sgRNA_R. A second round of PCR was used to add Illumina adaptor sequences using primers IlluminaAd_F and IlluminaAd_R. PCR primers are shown in *Supplementary file 5*. Purified PCR products were sequenced using next generation sequencing (NGS) at GeneWiz (South Plainfield, NJ). Fastq files were trimmed, aligned, and analyzed using PinAPL-Py (*Spahn et al., 2017*) and raw read counts were normalized as counts per million reads (cpm). The fold-depletion for each sgRNA was calculated for LNCaP or 22Rv1 relative to control human SLC knockout library. sgRNA rankings and significance (Sidak correction method) were determined according to the negative binomial model implemented in PinAPL-Py (*Spahn et al., 2017*). Gene-depletion significance was determined by a permutation test ($10^4$ permutations), and p values were corrected for multiple testing using the Sidak correction method.

## RNA-seq and network analyses

LNCaP cells were seeded into 6-well culture plates and treated with either vehicle (DMSO) or 2 µM GCN2iB in standard culture media as described above for 6 or 24 hr as indicated. Total RNA was extracted from cells using a RNeasy Plus Kit (Qiagen, Cat. #74034) according to the manufacturer's instructions. RNA quality control, library preparation, and NGS paired-end sequencing on Illumina HiSeq platform were carried out at GeneWiz (South Plainfield, NJ). Raw fastq data files were trimmed to remove adaptor sequences using cutadapt (*Martin, 2011*) and the quality of the trimmed reads was analyzed by FASTQC (*Andrews, 2010*) available in the Trim Galore package (*Krueger, 2012*). The trimmed paired sequencing reads were aligned to human reference genome GRCh38 downloaded from ENSEMBL using HISAT2 (*Kim et al., 2015*), resulting in concordant alignment of >92%.

FeatureCounts (*Liao et al., 2014*) from the Subread package was used to count the number of reads mapping to unique exonic sequences and was used for differential gene expression analysis using DESeq2 R-package (*Love et al., 2014*), comparing the DMSO-treated control group to the GCN2iB-treated groups.

The computer cluster Carbonate that was configured for high-performance analyses was used to perform all the steps of sequencing analysis from trimming to differential analysis. Volcano plots were created from differentially expressed genes using GraphPad Prism. Differentially expressed genes with a log2 fold change of ≥±1 and adjusted p values of ≤0.05 were considered significant for further analysis. GSEA (*Mootha et al., 2003*; *Subramanian et al., 2005*) using the Hallmark and Curated (C2) gene sets from the Molecular Signatures Database (*Liberzon et al., 2015*; *Liberzon et al., 2011*) was used to identify enriched pathways from lists of significant differentially expressed genes. The GCN2-dependent gene signature was derived from genes showing ≤−twofold change (fdr ≤0.05) following treatment with GCN2iB for 6 hr in our RNA-seq analysis. The RNA-seq data generated in this study have been deposited in the NCBI's Gene Expression Omnibus (GEO) database under the accession code GSE196252 [Reviewer's Token #kpkvcqcunpidhol].

## tRNA charging measurements

The levels of charged (aminoacylated) tRNAs genome-wide were measured using the CHARGE-seq method (*Pavlova et al., 2020*) with some protocol adjustments as previously described (*Misra et al., 2021*). The CHARGE-seq method relies on the simultaneous sequencing of both a charged and uncharged tRNA library. LNCaP cells were treated with vehicle (DMSO), GCN2iB, or GCN2iB in media supplemented with EAA for 8 hr. Total RNA was extracted using TRIzol (Life Technologies, Cat. #15596018) and a portion (2 µg) of total RNA for each sample was left untreated or oxidized by treatment with 12.5 mM $NaIO_4$ in sodium acetate solution (pH 4.5) in the dark at room temperature for 20 min followed by quenching with 0.3 M glucose. All mature tRNAs end with 3'-CCA and $NaIO_4$ will only oxidize the 3'-A residue of uncharged tRNAs since the 3'-ends of charged tRNAs are protected by linked amino acids. The resulting total RNA was desalted by passing through MicroSpin G-50 column (GE Healthcare, Cat. #27533001) and the tRNA was deacylated through β-elimination in 50 mM Tris–HCl (pH 9.0) at 37°C for 45 min followed by quenching with a solution containing 50 mM sodium acetate buffer (pH 4.5) and 100 mM NaCl. Total RNA was then recovered by overnight EtOH precipitation at −20°C. Following deacylation, libraries for NGS sequencing were prepared as follows, RNA samples were treated with polynucleotide kinase (New England BioLabs, Cat. #M0201) to remove the 3'-phosphate group from the uncharged tRNA followed by ligation to 5'-adenylated uniquely barcoded adapters using RNA ligase 2 truncated KQ (New England BioLabs, Cat. #M0351L). The resulting tRNAs were then ligated to a 5'-adaptor, annealed to an RT primer, and converted to cDNA using a using a SuperScript IV RT kit (Invitrogen, Cat. #18090050). The tRNA library was PCR amplified for 10 cycles using Illumina multiplex and barcode PCR primers and Platinum SuperFi II DNA Polymerase (Thermo Fisher Scientific, Cat. #12369002). All oligonucleotide primers and adapters utilized in the procedure are listed in *Supplementary file 5*. The resulting PCR products were electrophoresed on an 8% non-denaturing polyacrylamide TBE gel (Invitrogen, Cat. #EC6215) and ~210 bp bands were excised, purified, and subjected to Illumina HiSeq paired-end sequencing. Fastq files from the pooled samples were demultiplexed according to the five-nucleotide sample barcode contained within the ligated adapter using a custom python script (*Carlson, 2022*), and read pairs without an identifiable barcode were discarded. Illumina sequencing adapters and ligated sample barcode adapters were trimmed using cutadapt (*Martin, 2011*) in paired read mode and reads with less <20 nucleotides remaining after trimming, corresponding to empty adapters, were removed.

A custom reference file for human tRNA was constructed from the gtRNAdb GRCh38 high confidence tRNA gene set by removing duplicate sequences and appending 'CCA' to the 3' end of each unique tRNA gene sequence. Trimmed reads were then aligned to the custom tRNA reference file using bowtie2 (*Langmead and Salzberg, 2012*) short read aligner with options '--dovetail -D 20R 3N 1L 20 -i S,1,0.50'. The charging status of the aligned tRNA was determined from the 3' end of the aligned portion of each read pair, where reads ending in 'CCA' were determined to be charged and reads ending in 'CC' were determined to be uncharged. This method was able to determine the charging status of ~86% of all mapped reads. Counts of charged and uncharged tRNA for each tRNA gene in each sample were then imported to R for further analysis. Differences in read counts between

samples were accounted for by multiplying the read counts in each sample by a normalization coefficient equal to the average number of reads across all samples divided by the read count in that sample. The fraction charged for each tRNA isodecoder within each sample was determined by summing the normalized counts for charged and uncharged reads for each gene corresponding to that isodecoder, and the mean was then averaged for each of the four replicates for each treatment. The mean fraction charged ± standard deviation for each isodecoder was visualized using GraphPad Prism. Statistical significance in difference of the mean fraction charged was determined using a Welch's unpaired $t$-test with Benjamini–Hochberg fdr correction to account for repeated measures. The CHARGE-seq data generated in this study have been deposited in the NCBI's GEO database under the accession code GSE196251 [Reviewer's Token #kpkvcqcunpidhol].

Levels of total and aminoacylated tRNA[His] were also carried out by qRT-PCR as previously described (*Misra et al., 2021*) using primers specific for the GTG-tRNA[His] isodecoder tRNA[His]-FW and tRNA-[His]-REV (*Supplementary file 5*).

## Animal studies

All animal experiments were approved by the Institutional Animal Care and Use Committee (IACUC) at Indiana University School of Medicine (Protocol #21014) and comply with all regulations for ethical conduct of animal research. Male 4- to 6-week-old NSG (NOD.Cg-Prkdcscid Il2rgtm1Wjl/SzJ, RRID:IMSR_JAX:021885) mice were purchased from the In Vivo Therapeutics Core at Indiana University School of Medicine. GCN2iB used for animal studies was formulated in 0.5% methyl cellulose (Sigma-Aldrich, Cat. #M0262) containing 5% DMSO. Mice were dosed with vehicle or 30 mg/kg GCN2iB twice daily for 5 days/week by oral gavage as described in the legends to the figures. For LNCaP, 22Rv1, and PC-3 cell line-derived xenograft models, $2 \times 10^6$ cells in 100 μl of serum-free RPMI-1640 media containing 50% Matrigel (Corning, Cat. #356237) were injected subcutaneously into the dorsal flank of mice. Once the tumors reached a palpable stage (~50–100 mm³), the animals were randomized to different treatment groups using RandoMice v1.1.5 (*van Eenige et al., 2020*). The LuCaP-35 castration-resistant (LuCaP-35 CR) PCa PDX model (*Corey et al., 2003*; *Montgomery et al., 2008*) was received from Dr. Eva Corey (University of Washington) and were maintained in male castrated NSG mice for all experiments to conserve castration-resistant status. TM00298 PCa PDX model (*Sekhar et al., 2019*) was obtained from Jackson Laboratory and male NSG mice were engrafted with ~1 mm³ PDX tumor fragments and randomized once the tumors reached a palpable stage (~50–100 mm³). For all xenograft studies, tumor volume was monitored twice per week in a blinded fashion using a digital caliper and the formula: Tumor volume = (width² × length)/2. Body weight was determined once per week during the course of each study. At the end of the studies, mice were euthanized using $CO_2$ and tumors were harvested, weighed, and fixed for immunohistochemistry staining or flash frozen for immunoblot analyses. For the amino acid supplementation experiment in vivo, mice bearing 22Rv1 WT or 22Rv1 GCN2 KO tumors were supplemented with EAA (1.5 mg amino acids/g body weight) in the drinking water. The control group received drinking water with no amino acid supplementation. Supplementation of amino acids was started at the same time as tumor cell implantation and was maintained for the remainder of the study. Amino acids were dissolved in water in quantities determined by calculating average daily intake of drinking water for mice as previously reported (*Corsetti et al., 2014*). Composition, relative percentage, and an estimate of dietary intake of each amino acid from the drinking water are reported in *Supplementary file 5D*.

## Immunohistochemistry

Tumor tissue specimens were fixed for 24 hr in neutral buffered formalin, paraffin-embedded and sectioned (5 μm). Prostate core needle biopsy specimens were obtained from the Indiana University Comprehensive Cancer Center Tissue Procurement and Distribution Core and approval was granted by the Institutional Review Board (IRB #1796) at the Office of Research Administration at Indiana University. Human tumor tissue microarrays were purchased from US Biomax (Rockville, MD, Cat. #PR1921b and PR807c). Tissue sections were deparaffinized and rehydrated through serial graded xylene and alcohol washes. H&E staining was performed using standard methods. Antigen unmasking was achieved by boiling the slides in sodium citrate buffer (10 mM sodium citrate, 0.05% Tween-20, pH 6.0) for 10–15 min. Sections were further incubated in hydrogen peroxide to reduce endogenous activity. Tissue blocking was performed with 5% Horse Serum (Vector Laboratories Cat. #PK-7200) in

PBS and incubated with primary antibodies: Ki-67 (1:200, Abcam Cat. #ab16667, RRID:AB_302459), p-GCN2-T899 (1:100, Abcam Cat. #ab75836, RRID:AB_1310260), and SLC3A2 (4F2hc/CD98) (1:500, Cell Signaling Technology Cat. #47213, RRID:AB_2799323). After primary antibody incubation, tissue sections were washed, incubated in horseradish peroxidase-conjugated anti-rabbit secondary antibody according to the manufacturer's protocol (Vector Laboratories, Cat. #PK-7200), enzymatically developed in diaminobenzidine (DAB), and counterstained with hematoxylin. Tissue sections were then dehydrated and mounted with coverslips sealed with Cytoseal 60 mounting media (Thermo Fisher Scientific, Cat. #23-244257). Stained sections were scanned under brightfield using a Leica Aperio AT2 slide scanner (Leica Biosystems). Quantification was performed using Quantitative Pathology & Bioimage Analysis software (QuPath). Representative images were acquired using an EVOS M5000 cell imaging microscope (Life Technologies).

## Statistical analysis

All statistical tests were carried out using GraphPad Prism version 9.2.0 for Windows as indicated in the legends to the figures or as outlined in *Supplementary file 1*.

## Data availability statement

The authors declare that all data generated or analyzed in this study are included in the published article, its supplementary information and source files, or are publicly available. The CHARGE-seq and RNA-seq datasets generated in this study have been deposited in the NCBI GEO database under the ascension codes GSE196251 and GSE196252, respectively. The custom python script used in the analysis of our Charge-seq study is available on GitHub (https://github.com/carlsonkPhD/tRNA_Charge-Seq/, swh:1:rev:7705eb602023b45ef749b64a192d4cf1a2cdea36, *Carlson, 2022*). Gene expression data from prostate cancer patients (PRAD) in the TCGA database used for correlation analysis are publicly available.

## Acknowledgements

This work was supported in part by grants from NIH GM136331 (RCW), DK109714 (TGA, RCW), the NCI R21CA221942 (RP), the Indiana University Melvin and Bren Simon Comprehensive Cancer Center P30CA082709 (KAS), and the Lilly Endowment, Inc, through its support for the Indiana University Pervasive Technology Institute. We thank the IU Simon Comprehensive Cancer Center for use of the Tissue Procurement and Distribution Core, which provided core needle biopsy specimens for staining, the Preclinical Modeling and Therapeutics Core (PMTC) for assistance with flow cytometry and cell cycle analysis, and the Histology Lab Service Core at the Indiana University School of Medicine for embedding and sectioning of tissue samples. Finally, the authors acknowledge Ludovica Ceci for assistance with slide scanning, Kadir Isidan for assistance with histology, and members of Wek and Pili laboratories for helpful discussions.

## Additional information

### Competing interests

Tracy G Anthony: is a consultant for HiberCell, Inc. Ronald C Wek: is a member of the advisory board and holds equity in HiberCell, Inc. Kirk A Staschke: is a consultant for HiberCell, Inc and receives research support from HiberCell, Inc. The other authors declare that no competing interests exist.

### Funding

| Funder | Grant reference number | Author |
| --- | --- | --- |
| NIH Office of the Director | GM136331 | Ronald C Wek |
| NIH Office of the Director | DK109714 | Tracy G Anthony |
| National Cancer Institute | R21CA221942 | Roberto Pili |

| Funder | Grant reference number | Author |
| --- | --- | --- |
| Indiana University Melvin and Bren Simon Comprehensive Cancer Center | P30CA082709 | Kirk A Staschke |

The funders had no role in study design, data collection, and interpretation, or the decision to submit the work for publication.

## Author contributions

Ricardo A Cordova, Conceptualization, Data curation, Formal analysis, Validation, Investigation, Visualization, Methodology, Writing – original draft, Writing – review and editing, conceived of and designed the experiments; Jagannath Misra, Formal analysis, Investigation, Visualization, Methodology, Writing – original draft, carried out the CHARGE-seq and tRNA charging analysis; Parth H Amin, Data curation, Formal analysis, Investigation, Visualization, Methodology, Writing – original draft, provided bioinformatics support for the RNA-seq and CHARGE-seq studies; Anglea J Klunk, Data curation, Investigation, provided technical assistance with cell culture experiments and immunoblot analyses; Nur P Damayanti, Data curation, Investigation, provided technical advice and assisted with in vivo experiments; Kenneth R Carlson, Data curation, Formal analysis, Investigation, Visualization, Methodology, Writing – original draft, provided bioinformatics support for the RNA-seq and CHARGE-seq studies; Andrew J Elmendorf, Data curation, Investigation, provided technical assistance with cell culture experiments and immunoblot analyses; Hyeong-Geug Kim, Data curation, Investigation, Methodology, provided technical advice and assistance with IHC staining; Emily T Mirek, Investigation, Methodology, assisted with amino acid analysis; Bennet D Elzey, Resources, Investigation, provided technical advice and assistance with the LuCaP-35 CR animal model; Marcus J Miller, Data curation, Investigation, Methodology, Writing – original draft, assisted with amino acid analysis; X Charlie Dong, Investigation, Writing – review and editing, provided advice on IHC analysis and reviewed the manuscript; Liang Cheng, Formal analysis, Investigation, provided pathology interpretation of prostate cancer specimens; Tracy G Anthony, Conceptualization, Resources, Formal analysis, Investigation, Visualization, Methodology, Writing – original draft, Writing – review and editing, assisted with amino acid analysis; Roberto Pili, Conceptualization, Formal analysis, Supervision, Funding acquisition, Investigation, Visualization, Methodology, Writing – original draft, Project administration, Writing – review and editing, conceived of and designed the experiments; Ronald C Wek, Conceptualization, Resources, Data curation, Formal analysis, Supervision, Funding acquisition, Validation, Investigation, Visualization, Methodology, Writing – original draft, Project administration, Writing – review and editing, conceived of and designed the experiments; Kirk A Staschke, Conceptualization, Resources, Data curation, Formal analysis, Supervision, Funding acquisition, Validation, Investigation, Visualization, Methodology, Writing – original draft, Project administration, Writing – review and editing, conceived of and designed the experiments

## Author ORCIDs

Kirk A Staschke (ID) http://orcid.org/0000-0001-8722-9585

## Ethics

All animal experiments were approved by the Institutional Animal Care and Use Committee (IACUC) at Indiana University School of Medicine (Protocol #21014) and comply with all regulations for ethical conduct of animal research. Human prostate core needle biopsy specimens were obtained from the Indiana University Comprehensive Cancer Center Tissue Procurement and Distribution Core and approval was granted by the Institutional Review Board (IRB #1796) at the Office of Research Administration at Indiana University.

## Decision letter and Author response

Decision letter https://doi.org/10.7554/eLife.81083.sa1
Author response https://doi.org/10.7554/eLife.81083.sa2

# Additional files

## Supplementary files

- Supplementary file 1. Statistical analysis for growth assays.
- Supplementary file 2. Normalized read counts and analysis of RNA-seq data.
- Supplementary file 3. Normalized read counts and analysis of Charge-seq data.
- Supplementary file 4. CRISPR screen analysis of depleted genes and single guide RNAs (sgRNAs).
- Supplementary file 5. Supplementary tables.
- MDAR checklist
- Source data 1. Western blot source data for main figures, including all individual uncropped western blot images.
- Source data 2. Western blot source data for *Figure 1—figure supplement 1* and *Figure 1—figure supplement 2*, including all individual uncropped western blot images.
- Source data 3. Western blot source data for *Figure 1—figure supplement 3*, including all individual uncropped western blot images.
- Source data 4. Western blot source data for *Figure 1—figure supplement 5*, *Figure 3—figure supplement 2*, *Figure 3—figure supplement 4*, *Figure 4—figure supplement 3*, and *Figure 6—figure supplement 2*, including all individual uncropped western blot images.
- Source data 5. Western blot source data for *Figure 6—figure supplement 3*, *Figure 6—figure supplement 4*, and *Figure 7—figure supplement 2*, including all individual uncropped western blot images.

## Data availability

The authors declare that all data generated or analyzed in this study are included in the published article, its supplementary information and source files, or are publicly available. The CHARGE-seq and RNA-seq datasets generated in this study have been deposited in the NCBI Gene Expression Omnibus (GEO) database under the ascension codes GSE196251 and GSE196252, respectively. The custom python script used in the analysis of our Charge-seq study is available on GitHub (https://github.com/carlsonkPhD/tRNA_Charge-Seq/, copy archived at swh:1:rev:7705eb602023b45ef749b64a192d4c-f1a2cdea36). Gene expression data from prostate cancer patients (PRAD) in the TCGA database used for correlation analysis are publicly available.

The following datasets were generated:

| Author(s) | Year | Dataset title | Dataset URL | Database and Identifier |
|---|---|---|---|---|
| Cordova R, Misra J, Amin PH, Klunk AJ, Damayanti NP, Carlson KR, Elmendorf A, Kim H, Mirek ET, Elzey BD, Miller MJ, Dong X, Cheng L, Anthony TG, Pili R, Wek RC, Staschke KA | 2022 | GCN2 eIF2 kinase promotes prostate cancer by maintaining amino acid homeostasis | http://www.ncbi.nlm.nih.gov/geo/query/acc.cgi?acc=GSE196251 | NCBI Gene Expression Omnibus, GSE196251 |
| Cordova R, Misra J, Amin PH, Klunk AJ, Damayanti NP, Carlson KR, Elmendorf A, Kim H, Mirek ET, Elzey BD, Miller MJ, Dong X, Cheng L, Anthony TG, Pili R, Wek RC, Staschke KA | 2022 | GCN2 eIF2 kinase promotes prostate cancer by maintaining amino acid homeostasis | http://www.ncbi.nlm.nih.gov/geo/query/acc.cgi?acc=GSE196252 | NCBI Gene Expression Omnibus, GSE196252 |

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
