## [Editor Report]

This manuscript is an important body of work that addresses the role of the integrated stress response (ISR) and the role of the GCN2 protein kinase in prostate cancer. The studies comprehensively elucidate how GCN2 and amino acid transporters and uptake promote prostate cancer proliferation, as well as the therapeutic potential of inhibiting this pathway. This work, therefore, provides insights for both identification of new mechanisms and experimental therapeutics in prostate cancer.

---

## [Decision Letter]

**Decision letter after peer review:**

Thank you for submitting your article "GCN2 eIF2 kinase promotes prostate cancer by maintaining amino acid homeostasis" for consideration by *eLife*. Your article has been reviewed by 3 peer reviewers, and the evaluation has been overseen by a Reviewing Editor and David Ron as the Senior Editor. The following individual involved in the review of your submission has agreed to reveal their identity: Andrew C Hsieh (Reviewer #2).

Essential revisions:

1) Please address the analysis and role of p-eIF2 and other eIF2 kinases as commented on by Reviewer #1.

2) Please address the role of GCN2 inhibition in non-cancer prostate models, as commented by Reviewer #2.

3) There are a few issues related to presentation, formatting, and discussion raised by all reviewers. Please address these.

*Reviewer #1 (Recommendations for the authors):*

Cordova and colleagues provide very strong evidence for a tumorigenic role of GCN2 in prostate cancer via the upregulation of genes encoding for amino acid transporters like SLC3A2. Overall, this is a very interesting study with carefully performed experiments and high-quality results. An interesting question is whether GCN2 indeed exerts its effect via the induction of an ISR response, namely, upregulation of p-eIF2 and downstream effectors like ATF4. The data are devoid of analysis of p-eIF2 in the pancreatic cell lines in which GCN2, and other eIF2 kinases, are depleted by genetic means. It is of interest that p-eIF2 is marginally affected by the loss of GNC2 in tumors excised from nude mice (e.g. Figure S15). The authors need to explain this matter in the manuscript better (Discussion section).

Another issue is what activates GCN2 in prostate tumors. This is not explained very well in the manuscript. The authors need to provide some mechanistic explanation for the activation of GNC2 from their data.

An issue of potential concern is the specificity of GCN2i. Data in Figure 1D should be supplemented by the analysis of the proliferation of GCN2 KO tumor cells (CRISPR) in the presence of GCN2i.

The study would largely benefit from data showing ATF4 levels along p-GCN2 in human and/or mouse tumors from the xenograft assays by IHC.

*Reviewer #2 (Recommendations for the authors):*

Overall, I commend the authors for an excellent piece of work. It was a great read. My comments here are just to make the paper easier to read and provide more clarity on the specificity of the GCN2 axis in prostate cells.

Recommend conducting GCN2i or KD experiments in normal prostate epithelial cells to determine if their findings in prostate cancer are unique to cancer. Can use BPH1 or RWPE-1 cell lines for example.

The authors should combine their statistics (Supplementary File 1) into the figure legends or figures. Would highlight the most cogent statistics. It is unwieldy to read a paper and need to refer to another set of tables to confirm significance. This will greatly enhance the flow of the paper.

Line 166 – Sup Figure 10A is the wrong figure reference.

Given that there are a number of amino acids that are decreased upon GCN2 inhibition (Figure 3A), why do you think only His charging is affected in Figure 3D? Is it because there is only 1 isodecoder? The authors should discuss this potential specificity in the text. Figure 3D and Supplementary file 3 do not include any statistics. These should be included for each isoacceptor and isodecorder.

Line 177-179 – This sentence is not accurate at this point of the paper. The author does not show that cell grown without histidine leads to a decrease in p-GCN2-T899. This experiment is presented in 4D. Would edit accordingly.

Figure 4 – The font is tiny making the graph hard to read.

Supplementary Figure 19 – legend – LuCaP-34 should be LuCaP-35.

Administrative issue – the author state in the methods that the LuCa{-35CR line was a gift from Dr. Tim Ratliff. However, this line was generated at the University of Washington by Dr. Eva Corey, and no mention of permission or an MTA was mentioned in the manuscript.

*Reviewer #3 (Recommendations for the authors):*

Because of the extent of the experiments described, the manuscript is very long. In places, I felt that the figures could be improved to make the data more easily interpreted by readers and I had to read the text several times as the figures did not make it immediately clear what was being shown.

Examples where presentation changes could bring clarity are indicated below.

i) Adding titles or more descriptive plot axis legends to make it clearer what is being plotted against what without having to refer to the main text, eg In Figure 2A The fold change of what is being shown? Is this mean, median data, or one repeat? This applies to multiple figures eg also Figures 4A and 4B.

ii) In Figures 2C and 3A the color scheme is not as clear as it could be. This figure would be improved by converting to log2 space (as done for volcano plots) and having distinct divergent increasingly darker shades for + and – values. As presented a fold change of 1 is no change at all but has a rusty-orange hue when having this the palest shade or white would seem more appropriate. Also a 2-fold reduction or greater (ie <0.5) are all the same black color in the current figure. There are 4 squares for each treatment, are these individual repeats? How are these data normalised? Would it not be more typical to present a log2 of the ratio of GCN2iB/vehicle for the 2 time points.

iii) The remarkable specificity of the effect on tRNAHis-GUG is rather lost in Figure 3D. It might help just showing this tRNA here and making the rest of the plot larger as a supplement.

iv) It is not clear why there appears a random use of 4F2 vs SLA3A2, especially in Figure 4. Having a single consistent term will be easier for readers, especially those not versed in the naming history here.

Other points

1. Western blot quantifications. For several blot figures, it is not clear how many repeats were done. Are some n=1? For Figure 1B lines 67-8 state GCN2 depletion lowered p-eIF2 levels, however, it is not that clear from the blot image shown and there is no quantification in the supplementary file.

2. Line 71. Can it be clarified why different siRNAs were used to knock down GCN2 in LAPC4 cells?

3. Can the authors comment on the apparent wide variations in ISR responses to low doses of GCN2iB in the different cell lines (Supp. Figure 2B) which does not correlate that well with the impact on cell growth.

4. 4. Line 185. Should 'restored' be 'reversed'?

5. Line 357. Given the varied diets people consume, it may not be likely targeting GCN2 alone will be 'a potent strategy'. It seems more likely that inhibition of GCN2 in combination with a second target might be a better strategy. Ie perhaps say that targeting GCN2 may contribute to a potent strategy.

6. Given the impact of histidine in the cell models, are the expression-impacted transporter genes particularly enriched for histidine codons?

7. I was surprised the discussion did not include a reference to mTOR. In other studies, mTOR has been linked to amino acid uptake and to GCN2 and mTOR inhibitors have been included in a range of cancer therapy studies.

---

## [Author Response]

Reviewer #1 (Recommendations for the authors):Cordova and colleagues provide very strong evidence for a tumorigenic role of GCN2 in prostate cancer via the upregulation of genes encoding for amino acid transporters like SLC3A2. Overall, this is a very interesting study with carefully performed experiments and high-quality results. An interesting question is whether GCN2 indeed exerts its effect via the induction of an ISR response, namely, upregulation of p-eIF2 and downstream effectors like ATF4. The data are devoid of analysis of p-eIF2 in the pancreatic cell lines in which GCN2, and other eIF2 kinases, are depleted by genetic means. It is of interest that p-eIF2 is marginally affected by the loss of GNC2 in tumors excised from nude mice (e.g. Figure S15). The authors need to explain this matter in the manuscript better (Discussion section).

This concern involves the addition of p-eIF2α and total eIF2α measurements in our analysis of cultured parental (LNCaP, 22Rv1, and PC-3) and GCN2 KO (22Rv1 and PC-3) PCa cells that are featured in Figure 1 —figure supplement 1A and Figure 1 —figure supplement 2A. We carried out these additional immunoblot experiments and now show that loss of GCN2 reduces p-eIF2α and ATF4 levels using two independent siRNAs (Figure 1B) or following CRISPR KO (Figure 1 —figure supplement 2A). The p-eIF2α and ATF4 levels were not reduced with the knock-down of the other eIF2α kinases (Figure 1 —figure supplement 1A).

Additionally, there was concern about the observation that p-eIF2α levels were not appreciably reduced in the GCN2 KO-derived tumors prepared from xenografted mice. We do note that downstream targets of GCN2 and p-eIF2α, such as 4F2, LAT1, xCT, ASCT1, ASCT2, and CAT1 proteins, were sharply lowered with genetic loss of GCN2 (Figure 6B and D, Figure 6 —figure supplement 2A, and Figure 6 —figure supplement 3C).

These are end-stage tumor preparations in which tumors were isolated after 42 days of growth. In our in vitro studies, we have observed induction of p-eIF2α in the absence of GCN2 at later times during nutrient limiting conditions. However, this p-eIF2α is not accompanied by the downstream induction of the ISR. It appears that either a secondary eIF2α kinase is activated, or there is a reduction of phosphatase-directed dephosphorylation of eIF2α, which could account for observed p-eIF2α in the absence of GCN2. The p-eIF2α occurs late in the timing of the ISR induction and is viewed as a consequence of secondary stresses or regulatory events imposed by the primary stress. The onset of p-eIF2α in the absence of the primary eIF2α kinase at later times of stress has been reported earlier (e.g. PMID: 12215525 and 14729979). Therefore, with these late-stage preparations, we view the downstream ISR effectors as being the best measure of GCN2-directed response. We have briefly added these points to the Results section on page 14, lines 307-310.

Another issue is what activates GCN2 in prostate tumors. This is not explained very well in the manuscript. The authors need to provide some mechanistic explanation for the activation of GNC2 from their data.

Our results show that in vitro PCa cells are limited for essential amino acids. In the case of LNCaP cells, histidine is limiting, and there are measureable increases in unchaged tRNA^His^. The addition of essential amino acids to the culture media reduced GCN2 activation and the ISR, consistent with the idea that limiting amino acids is an activator of GCN2 in PCa cells cultured in standard growth medium. In tumors, we know that there is also a reduction in many free amino acids, and the addition of essential amino acids to the drinking water partially rescued growth of the GCN2 KO PCa tumors, but had no effect on PCa with functional GCN2. Therefore, our in vitro and in vivo studies suggest that PCa profileration induces sufficient amino acid limitation to activate GCN2, and this activation is required to sustain growth. We cover these points on page 18, lines 403– 410 of the Discussion.

An issue of potential concern is the specificity of GCN2i. Data in Figure 1D should be supplemented by the analysis of the proliferation of GCN2 KO tumor cells (CRISPR) in the presence of GCN2i.

To further address the specificity of Gcn2iB, we treated parental or GCN2 KO 22Rv1 cells with 2 – 10 µM GCN2iB for up to 6 days. Consistent with our previous results shown in Figure 1 - Figure Supplement 3A, GCN2iB inhibited the growth of parental 22Rv1 cells. However, GCN2iB had no significant effect on the growth of GCN2 KO cells. These results demonstrate the selective inhibition of GCN2-dependent growth by GCN2iB at the concentrations used in our PCa studies.

The study would largely benefit from data showing ATF4 levels along p-GCN2 in human and/or mouse tumors from the xenograft assays by IHC.

We have added IHC staining for p-GCN2 in parental tumors, GCN2 KO tumors, and for tumors from mice treated with GCN2iB. We now show enhanced staining for p-GCN2 in parental tumors as compared to GCN2 KO tumors (22Rv1 and PC-3) or in tumors from GCN2iB treated mice (LNCaP, 22Rv1, TM00298, and LuCaP-35CR), and these results have been added to Figure 6 – Figure Supplement 1A (22Rv1 GCN2 KO tumors), Figure 6 —figure supplement 1C (PC-3 GCN2 KO tumors), and Figure 7 —figure supplement 1A (GCN2iB-treated mice bearing LNCaP, 22Rv1, TM00298, or LuCaP-35CR tumors). These new results are consistent with our previous immunoblot experiments showing that loss of GCN2 results in reduced expression of the GCN2-dependent target gene 4F2 (SLC3A2) in vivo (Figure 6B, Figure 6D, and Figure 6 – Figure Supplement 2A). In addition, we now show reduced IHC staining for 4F2 (SLC3A2) in GCN2 KO tumors as compared to parental controls (Figure 6 —figure supplement 1A and 1C).

We carried out IHC studies for ATF4 using two different commercial antibodies and our custom antibody for ATF4, and unfortunately did not measure specific staining as judged by staining of parental versus ATF4 KO tumors. ATF4 is expressed at low concentrations, as are many transcription factors, and this may be a contributor to the difficulties we encountered in these experiments. As noted above, other downstream target of GCN2, such as 4F2 and other amino acid transporters were reduced as judged by Western blotting in the GCN2 KO tumors.

Reviewer #2 (Recommendations for the authors):Overall, I commend the authors for an excellent piece of work. It was a great read. My comments here are just to make the paper easier to read and provide more clarity on the specificity of the GCN2 axis in prostate cells.Recommend conducting GCN2i or KD experiments in normal prostate epithelial cells to determine if their findings in prostate cancer are unique to cancer. Can use BPH1 or RWPE-1 cell lines for example.

We have added new experimental data for the non-tumorigenic prostate cell line BPH-1, and now show that GCN2 is not activated (as measured by p-GCN2 levels) in these cells as compared to PCa cell lines. In addition, we show that knock-down of GCN2, ATF4, or 4F2 (SLC3A2), or treatment with GCN2iB has no effect on the growth of non-tumorigenic prostate cells. These new results were added as Figure 1 —figure supplement 5 and the statistical analysis added to Supplementary File 1_StatisticalAnalysis.

The authors should combine their statistics (Supplementary File 1) into the figure legends or figures. Would highlight the most cogent statistics. It is unwieldy to read a paper and need to refer to another set of tables to confirm significance. This will greatly enhance the flow of the paper.

We added color-coded indicators of statistical significance to the last day for each of our growth assays in Figures 1A, 1C, 1D, 3B, 4F, 4H, 6G, and 6I and Figure 1 —figure supplements 2B, 3A, 3B, 4, and 5C, Figure 3 —figure supplement 2, Figure 4 —figure supplement 3D, and Figure 6 —figure supplement 3A. The reader may also refer to Supplementary File 1_StatisticalAnalysis for additional detail. We agree that these changes improve the rigor and readability of the manuscript.

Line 166 – Sup Figure 10A is the wrong figure reference.

We corrected this error, and by adding a new Supplemental Figure 5 and renumbering and renaming of subsequent figures, the reference to this figure is now Figure 3 —figure supplement 3A.

Given that there are a number of amino acids that are decreased upon GCN2 inhibition (Figure 3A), why do you think only His charging is affected in Figure 3D? Is it because there is only 1 isodecoder? The authors should discuss this potential specificity in the text. Figure 3D and Supplementary file 3 do not include any statistics. These should be included for each isoacceptor and isodecorder.

The reviewer correctly points out that only one of the two isodecoders for histidine was detected in our charge-seq analysis in LNCaP cells, and we have shown that these cells are limited for histidine. The second isodecoder was measureable, but expressed at low levels. At this time, we do not know whether this is due to differential expression of the two histine tRNA isocoders or is a consequence of the tRNA charge-seq method, which involves reverse transcription and PCR amplification for each of the tRNAs. While we are confident of the ratio of the tRNA charging measurements using the tRNA charge-seq method, we are cautious about determining concentration differences between tRNA isodecoders. We have observed reductions in HIS-GTG tRNA charging using both the bulk library charge-seq method (Figure 3D and E) and dedicated qRT-PCR methods (Figure 3 —figure supplement 4). We have added statistical analysis of the charge-seq data to Supplementary File 3_ChargeseqData.

Line 177-179 – This sentence is not accurate at this point of the paper. The author does not show that cell grown without histidine leads to a decrease in p-GCN2-T899. This experiment is presented in 4D. Would edit accordingly.

The sentence on line 177-179 of the manuscript now reads: "These results suggest that histidine depletion and the accompanying accumulation of uncharged tRNA_His_ is a key signal for activation of GCN2". In Figure 3A, we show that treatment with GCN2iB results in sharp reductions in amino acid levels, including histidine. In Figure 3B and Figure 3 —figure supplements 1 and 2, we show that addition of essential amino acids (including histidine) reverses the proliferation defect induced by GCN2iB treatment and restores amino acid pool levels. We further show that loss of GCN2 results in activation of the cell cycle checkpoint and G1 arrest, which was reversed by the addition of essential amino acids, including histidine (Figure 3C and Figure 3 —figure supplement 3A).

Figure 4 – The font is tiny making the graph hard to read.

We increased the font size for Figures 4A and 4B to improve the manuscript's readability.

Supplementary Figure 19 – legend – LuCaP-34 should be LuCaP-35.

We corrected this error in the legend to Figure 7 —figure supplement 1.

Administrative issue – the author state in the methods that the LuCa{-35CR line was a gift from Dr. Tim Ratliff. However, this line was generated at the University of Washington by Dr. Eva Corey, and no mention of permission or an MTA was mentioned in the manuscript.

Experiments using the LuCap-35 CR model were performed as part of a collaboration between Dr. Roberto Pili and Dr. Ben Elzey at the Purdue University Center for Cancer Research, who obtained the model from Dr. Eva Corey at the University of Washington via an MTA. Dr. Elzey has been added as an author to recognize his contribution to this experiment and Dr. Corey has now been appropriately acknowledged in the Materials and methods section of the manuscript. We apologize for this oversight.

Reviewer #3 (Recommendations for the authors):Because of the extent of the experiments described, the manuscript is very long. In places, I felt that the figures could be improved to make the data more easily interpreted by readers and I had to read the text several times as the figures did not make it immediately clear what was being shown.Examples where presentation changes could bring clarity are indicated below.i) Adding titles or more descriptive plot axis legends to make it clearer what is being plotted against what without having to refer to the main text, eg In Figure 2A The fold change of what is being shown? Is this mean, median data, or one repeat? This applies to multiple figures eg also Figures 4A and 4B.

The legend to Figure 2A states that the standard volcano plot represents log2 fold change comparing LNCaP cells treated with GCN2iB versus vehicle control for 24 hours. We now added a title to the volcano plot in Figure 2A to reflect this detail.

ii) In Figures 2C and 3A the color scheme is not as clear as it could be. This figure would be improved by converting to log2 space (as done for volcano plots) and having distinct divergent increasingly darker shades for + and – values. As presented a fold change of 1 is no change at all but has a rusty-orange hue when having this the palest shade or white would seem more appropriate. Also a 2-fold reduction or greater (ie <0.5) are all the same black color in the current figure. There are 4 squares for each treatment, are these individual repeats? How are these data normalised? Would it not be more typical to present a log2 of the ratio of GCN2iB/vehicle for the 2 time points.

The heatmap in Figure 2C is shown as fold change to emphasize the differences in *SLC* gene expression between untreated (vehicle) control cells and cells treated with GCN2iB for 6 or 24 hours. The data are normalized to the vehicle control samples, and each column represents an individual sample (N = 4 samples per treatment group). The data represented in Figure 2C have been added to Supplementary File 2_RNAseqData to allow for a more detailed analysis by the reader.

iii) The remarkable specificity of the effect on tRNAHis-GUG is rather lost in Figure 3D. It might help just showing this tRNA here and making the rest of the plot larger as a supplement.

We added a red box to further highlight the His-GTG isodecoder in the plot in Figure 3D, which is also highlighted in red.

iv) It is not clear why there appears a random use of 4F2 vs SLA3A2, especially in Figure 4. Having a single consistent term will be easier for readers, especially those not versed in the naming history here.

We modified the text in the manuscript to consistently refer to SLC3A2 as 4F2.

Other points1. Western blot quantifications. For several blot figures, it is not clear how many repeats were done. Are some n=1? For Figure 1B lines 67-8 state GCN2 depletion lowered p-eIF2 levels, however, it is not that clear from the blot image shown and there is no quantification in the supplementary file.

Each experiment has been performed at least twice, with two independent siRNAs or two independent GCN2 KO clones and we have added this information to the Materials and methods section and the figure legends. Furthermore, we now include quantitation of key immunoblot panels that pertain to the analysis of the GCN2 knock-downs or knockouts in PCa cells in Figures 1B and 1E, and Figure 1 —figure supplement 2A.

2. Line 71. Can it be clarified why different siRNAs were used to knock down GCN2 in LAPC4 cells?

The reason for the use of different siRNAs to reduce the expression of GCN2 in LAPC-4 cells pertains to technical difficulties we encountered getting efficient transfection efficiencies with standard siRNAs. We were not able to obtain good siRNA transfection of LAPC-4 cells with standard Lipofectamine RNAiMax or DharmaFECT 3; however, we successfully used Accell siRNA in Accell Delivery Media to achieve efficient transfection and knock-down of GCN2 expression. This technical point is included in the Materials and methods section of the manuscript.

3. Can the authors comment on the apparent wide variations in ISR responses to low doses of GCN2iB in the different cell lines (Supp. Figure 2B) which does not correlate that well with the impact on cell growth.

We noted variation in the sensitivity of different cell models to GCN2iB inhibition, which generally correlates with the ability of the inhibitor to reduce p-eIF2α levels. For example, LNCaP cells show a more robust inhibtion of p-eIF2 and ATF4 levels at low concentrations of GCN2iB as compared to 22Rv1 cells, and this correlates with increased sensitivity to GCN2iB (see Figures 1D and Figure 1 —figure supplements 3A and 3B).

4. 4. Line 185. Should 'restored' be 'reversed'?

We corrected the manuscript to read "only the addition of histidine to the culture medium reversed the growth inhibition triggered by GCN2iB".

5. Line 357. Given the varied diets people consume, it may not be likely targeting GCN2 alone will be 'a potent strategy'. It seems more likely that inhibition of GCN2 in combination with a second target might be a better strategy. Ie perhaps say that targeting GCN2 may contribute to a potent strategy.

We adjusted language pertaining to this comment at the end of our discussion. The sentence now reads as follows: "The development of new therapeutic strategies that work independently from inhibition of the AR axis is highly clinically relevant, and based on our findings, inhibitors of GCN2 should be considered further as part of a strategy for treatment of androgen-sensitive and castration-resistant prostate cancers".

6. Given the impact of histidine in the cell models, are the expression-impacted transporter genes particularly enriched for histidine codons?

We have not observed enrichment for histidine codons in the ORFs of genes encoding transporters affected by GCN2 such as *SLC3A2, SLC1A4, SLC1A5, SLC7A1, SLC7A5, or SLC7A11* (Avg histidine codon usage ~1.25%) as compared to the average human gene (Avg histidine codon usage ~1.3%; www.kazusa.or.jp/codon/). As noted in the manuscript, other models (i.e., 22Rv1) are limited for a different essential amino acid, that being lysine. Thus, the genetic background and unique metabolic profile of each model likely determines which amino acid or amino acids become limiting.

7. I was surprised the discussion did not include a reference to mTOR. In other studies, mTOR has been linked to amino acid uptake and to GCN2 and mTOR inhibitors have been included in a range of cancer therapy studies.

We observed a transient decrease in mTORC1 activity commensurate with decreased amino acids following inhibition of GCN2, and we now include a reference pertaining to mTORC1 and its importance in nutrient regulation of translation, and noted these findings in the Discussion section on page 20, lines 444 – 446.